# mTORC1/S6K1 signaling promotes sustained oncogenic translation through modulating CRL3[IBTK]-mediated ubiquitination of eIF4A1 in cancer cells

Dongyue Jiao[1†], Huiru Sun[1†], Xiaying Zhao[1], Yingji Chen[1], Zeheng Lv[2,3], Qing Shi[1], Yao Li[1], Chenji Wang[1]*, Kun Gao[2,3]*

[1]State Key Laboratory of Genetic Engineering, Shanghai Stomatological Hospital & School of Stomatology, MOE Engineering Research Center of Gene Technology, Shanghai Engineering Research Center of Industrial Microorganisms, School of Life Sciences, Fudan University, Shanghai, China; [2]Department of Clinical Laboratory, Shanghai First Maternity and Infant Hospital, School of Medicine, Tongji University, Shanghai, China; [3]Shanghai Key Laboratory of Maternal Fetal Medicine, Shanghai Institute of Maternal-Fetal Medicine and Gynecologic Oncology, Shanghai First Maternity and Infant Hospital, School of Medicine, Tongji University, Shanghai, China

*For correspondence:
Chenjiwang@fudan.edu.cn (CW);
kungao@tongji.edu.cn (KG)

[†]These authors contributed equally to this work

Competing interest: The authors declare that no competing interests exist.

**Abstract** Enhanced protein synthesis is a crucial molecular mechanism that allows cancer cells to survive, proliferate, metastasize, and develop resistance to anti-cancer treatments, and often arises as a consequence of increased signaling flux channeled to mRNA-bearing eukaryotic initiation factor 4F (eIF4F). However, the post-translational regulation of eIF4A1, an ATP-dependent RNA helicase and subunit of the eIF4F complex, is still poorly understood. Here, we demonstrate that IBTK, a substrate-binding adaptor of the Cullin 3-RING ubiquitin ligase (CRL3) complex, interacts with eIF4A1. The non-degradative ubiquitination of eIF4A1 catalyzed by the CRL3[IBTK] complex promotes cap-dependent translational initiation, nascent protein synthesis, oncogene expression, and cervical tumor cell growth both in vivo and in vitro. Moreover, we show that mTORC1 and S6K1, two key regulators of protein synthesis, directly phosphorylate IBTK to augment eIF4A1 ubiquitination and sustained oncogenic translation. This link between the CRL3[IBTK] complex and the mTORC1/S6K1 signaling pathway, which is frequently dysregulated in cancer, represents a promising target for anti-cancer therapies.

## eLife assessment

This study reports a novel substrate and a mediator of oncogenesis downstream of mTORC1, a **fundamental** advance in our understanding of the mechanistic basis of mTORC1-regulated cap-dependent translation and protein synthesis. Using an array of biochemical, proteomic and functional assays, the authors provide **compelling** evidence for a novel mTORC1/S6K1-IBTK-eIF4A1 signaling axis that promotes cancer pathogenic translation. This work is of broad interest and significance, given the importance of aberrant protein synthesis in cancer.

## Introduction

Enhanced protein synthesis is a critical process that allows cancer cells to survive, multiply, metastasize, and resist anti-cancer treatments (*Bhat et al., 2015*; *Fabbri et al., 2021*). This process typically

results from increased signaling through the eukaryotic initiation factor 4F (eIF4F), which binds to mRNAs. eIF4F complex is composed of four proteins: eIF4E, eIF4G, eIF4B, and eIF4A. Of these, eIF4A is responsible for unwinding the secondary structure in the 5'-UTR region of the mRNA, allowing the ribosomal subunit to bind and begin translation (*Merrick, 2015*). The eIF4A protein family consists of three paralogs, eIF4A1, eIF4A2, and eIF4A3. Although eIF4A1 and eIF4A2 have similar roles in translation initiation, eIF4A3 is not involved in translation control but an important component of the exon junction complex (EJC) and plays a critical role in nonsense-mediated mRNA decay (*Lu et al., 2014*). Increased eIF4F activity promotes the translation of mRNAs involved in cell proliferation and survival, and tumor immune evasion, which are hallmarks of cancer (*Boussemart et al., 2014*; *Cerezo et al., 2018*; *Malka-Mahieu et al., 2017*; *Pelletier et al., 2015*). Several eIF4A inhibitors, such as eFT226, silvestrol, hippuristanol, and pateamine A, have demonstrated promising anti-tumor activities in various cancer types conducted in vitro and in vivo studies (*Boussemart et al., 2014*; *Malka-Mahieu et al., 2017*; *Pelletier et al., 2015*). In particular, eFT226 is currently being evaluated in multiple clinical trials (*Ernst et al., 2020*).

The mTORC1 signaling pathway plays a crucial role in promoting cell growth and anabolism while inhibiting catabolism. It integrates various signals from nutrients, cellular energy levels, growth factors, and environmental stimuli to regulate this process. Translation is a highly energy-intensive process and tightly regulated by the mTORC1 signaling. Ribosomal protein S6 kinases (S6Ks) and 4E-binding proteins (4E-BPs) are two major downstream targets of the mTORC1 signaling. They regulate various aspects of mRNA translation in a phosphorylation-dependent manner. For example, S6Ks phosphorylate ribosomal protein S6, which promotes translation initiation and elongation. 4E-BPs, on the other hand, bind to eIF4E and inhibit its activity, preventing the translation of specific mRNAs. In addition, mTORC1 coordinates translation by phosphorylating other components of the translational machinery, such as eEF2K, eIF2B, and LARP1 (*Battaglioni et al., 2022*).

The Cullin 3-RING ubiquitin ligase (CRL3) complex subfamily is composed of a catalytic core made up of RBX1 and CUL3, and a substrate-binding adaptor that contains an interchangeable BTB domain. The CRL3 complex plays roles in regulating various cellular processes,such as cell division, differentiation, and signal transduction. Through its E3 ubiquitin ligase activity, the CRL3 complex promotes the degradation of specific proteins by marking them with ubiquitin. However, the CRL3 complex can also facilitate non-degradative ubiquitination, which can modulate substrate activity, localization, or interaction (*Genschik et al., 2013*). Initially characterized as an inhibitor of Bruton's tyrosine kinase (IBTK), the IBTK contains two BTB domains and can assemble an active CRL3 complex (*Liu et al., 2001*; *Pisano et al., 2015*). The downstream targets and biological processes of the CRL3[IBTK] complex are still poorly understood, but a limited number of studies suggest that IBTK plays a pro-survival role in cancer cells. IBTK was preferentially translated in response to eIF2α phosphorylation during endoplasmic reticulum stress. Knockdown (KD) of IBTK expression reduced cell viability and increased apoptosis in cultured cells (*Baird et al., 2014*; *Willy et al., 2017*). A genome-wide RNAi screen identified IBTK as a synthetic lethal partner of the Ras oncogene in colorectal cancer cells (*Luo et al., 2009*). In Eμ-myc transgenic mice, *Ibtk* gene knockout (KO) delayed the onset of pre-B/B lymphoma and improved animal survival due to increased apoptosis of pre-cancerous B cells (*Vecchio et al., 2019*). The expression of IBTK was significantly increased in chronic lymphocytic leukemia (CLL) progression, but decreased in remission following chemotherapy. KD of IBTK expression increased spontaneous and chemotherapy agent-induced apoptosis and impaired cell cycle progression in CLL cell lines (*Albano et al., 2018*). To date, only two substrates of the CRL3[IBTK] complex have been reported. CRL3[IBTK] targeted PDCD4, an eIF4A1 inhibitor, for proteasomal degradation (*Pisano et al., 2015*). Another study showed that CRL3[IBTK] activated β-catenin-dependent transcription of MYC by promoting GSK-3β degradation in cancerous B cells (*Vecchio et al., 2022*). These findings indicate that CRL3[IBTK] may play a crucial role in cancer development and progression, as well as in the regulation of cell survival and apoptosis.

Through a variety of biochemical and cell biology approaches, we have identified eIF4A1 as a non-degradative ubiquitination substrate of the CRL3[IBTK] complex. Our findings also suggest that a mTORC1/S6K1-IBTK-eIF4A1 signaling axis mediates cap-dependent translation, facilitates oncoprotein expression, and promotes tumor cell growth.

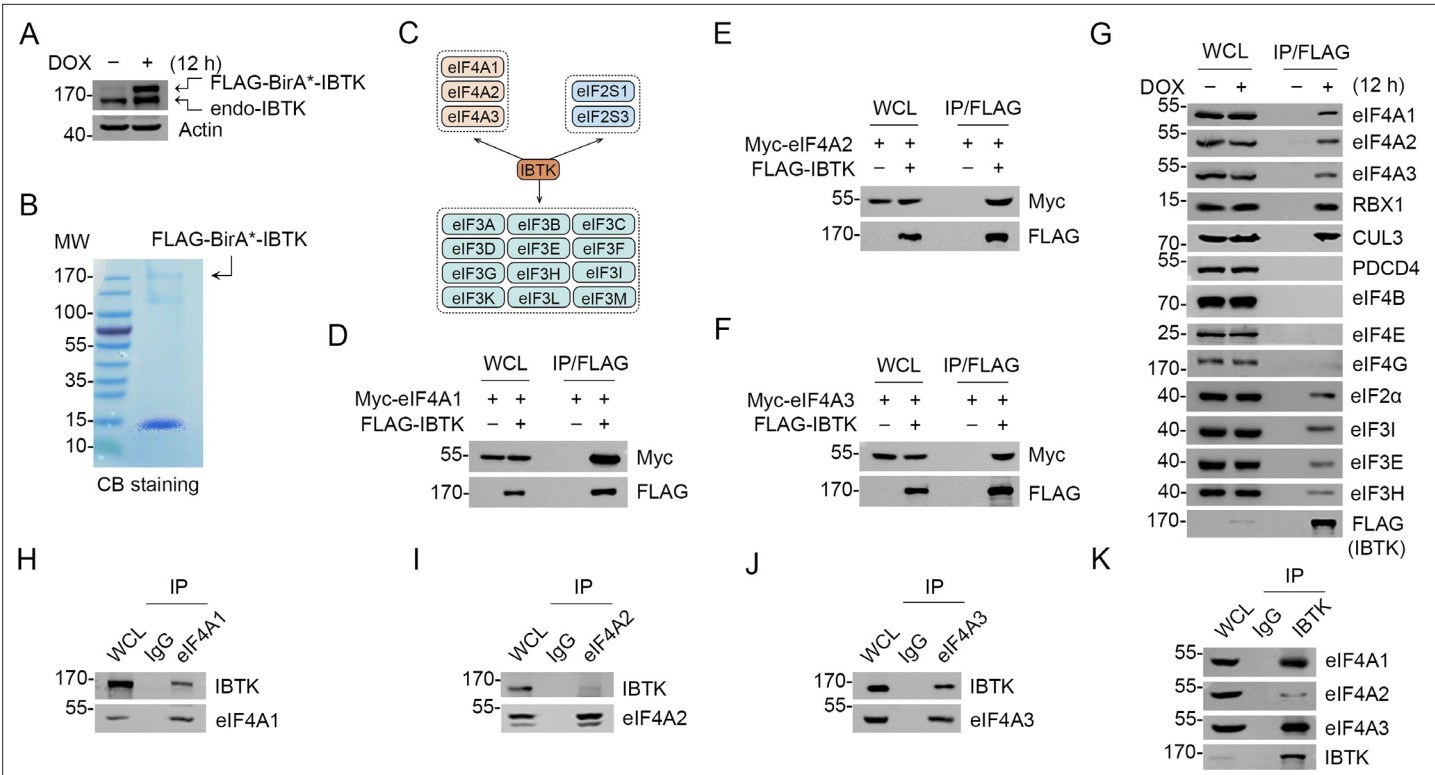

**Figure 1.** Inhibitor of Bruton's tyrosine kinase (IBTK) interacts with eIF4A1 in cells. (**A**) Western blot (WB) analysis of the indicated proteins in the whole cell lysates (WCLs) from FLAG-BirA*-IBTK Tet-on-inducible Flp-In T-REx 293 cells treated with doxycycline (DOX) (10 ng/ml) for 12 hr. (**B**) Affinity purification of IBTK-containing protein complexes using FLAG-M2 beads was conducted in FLAG-BirA*-IBTK Tet-on-inducible Flp-In T-REx 293 cells. The associated proteins were separated by SDS-PAGE and visualized by Coomassie Blue (CB) staining. (**C**) The eukaryotic initiation factor (eIF) interactome of IBTK identified by affinity purification coupled with mass spectrometry (AP-MS) analysis in (**B**). The full list of IBTK-binding partners was shown in *Supplementary file 1a*. (**D–F**) WB analysis of the indicated proteins in the WCLs and co-immunoprecipitation (co-IP) samples of anti-FLAG antibody obtained from 293T cells transfected with the indicated plasmids. (**G**) FLAG-BirA*-IBTK Tet-on-inducible Flp-In T-REx 293 cells were treated with/without DOX (10 ng/ml) for 12 hr, then collected and subjected to co-IP with anti-FLAG antibody. The WCLs and co-IP samples were prepared for WB analysis with the indicated antibodies. (**H–K**) Co-IP using anti-eIF4A1 (**H**), eIF4A2 (**I**), eIF4A3 (**J**), or IBTK (**K**) antibody in the WCLs prepared from 293T cells, followed by WB analysis with the indicated antibodies.

The online version of this article includes the following source data and figure supplement(s) for figure 1:

**Source data 1.** Original file for the western blot analysis in *Figure 1*.

**Source data 2.** Labeled file for the western blot analysis in *Figure 1*.

**Figure supplement 1.** Inhibitor of Bruton's tyrosine kinase (IBTK) interacts with eIF4A1 in cells.

## Results

### Identification of eIF4A family proteins as IBTK interacting proteins

Affinity purification coupled with mass spectrometry (AP-MS) and proximity-dependent biotinylation identification (BioID) methods are two powerful tools for interrogating protein-protein interactions (*Liu et al., 2018*). Using these techniques, we sought to comprehensively characterize the interaction partners of IBTK in living cells. To accomplish this, we developed a tag workflow that enabled simultaneous AP-MS and BioID analysis with a single construct, pcDNA5/FRT vector containing FLAG-BirA*-IBTK. This construct was stably transfected into Flp-In T-REx 293 cells, resulting in inducible expression of IBTK (*Figure 1—figure supplement 1A*). We observed that levels of doxycycline (DOX)-induced FLAG-BirA*-IBTK protein were comparable to those of endogenous IBTK (*Figure 1A*). Through our AP-MS and BioID analysis, we identified multiple potential IBTK interactors (*Supplementary file 1a and b*). One notable finding was the presence of multiple eIFs in the purified IBTK complexes, including the subunits of eIF2, eIF3, and eIF4 complex (*Figure 1B and C*, *Figure 1—figure supplement 1B*). Of particular interest, eIF4A1 is a crucial subunit of the eIF4F complex and a promising

therapeutic target for cancer. Previous large-scale interactome mapping datasets have also revealed a potential interaction between IBTK and eIF4A1, though it had not been analyzed for its biological functions (*Huttlin et al., 2021*). Therefore, we performed additional analyses to explore the pathophysiological significance of the eIF4A1-IBTK interaction.

To validate the interaction between IBTK and eIF4A1, we performed co-immunoprecipitation (co-IP) assays. Our results demonstrated that ectopically expressed IBTK and eIF4A1 interacted with each other (*Figure 1D*). Moreover, two eIF4A1 paralogs, eIF4A2 and eIF4A3, also interacted with IBTK (*Figure 1E and F*). Using FLAG-IBTK as an immunoprecipitation agent, we were able to confirm that endogenous eIF4A1/2/3 and CRL3 subunits (RBX1 and CUL3) associated with IBTK, but not other eIF4 complex subunits (eIF4B, eIF4E, and eIF4G) (*Figure 1G*). Additionally, we observed endogenous interactions between IBTK and eIF4A1/2/3 (*Figure 1H–K*).

Collectively, these data indicate that IBTK specifically interacts with three members of the eIF4A family in cells.

## CRL3[IBTK] mediates non-degradative ubiquitination of eIF4A1

Previous studies have demonstrated that the protein stability of eIF4A1 can be regulated by the ubiquitin-proteasome pathway, and that the deubiquitinase USP9X promotes eIF4A1 stability by facilitating its deubiquitination (*Li et al., 2018*). However, the specific E3 ubiquitin ligase(s) responsible for eIF4A1 proteolysis remain(s) elusive. Given that IBTK is engaged in the assembly of a CRL3 ubiquitin ligase complex, we hypothesized that IBTK may target eIF4As for degradation. Nevertheless, exogenous overexpression of IBTK did not influence the protein levels of endogenous eIF4A1/2/3 (*Figure 2A*). Moreover, CRISPR/Cas9-mediated KO (*Figure 2—figure supplement 1A-C*) or shRNA-mediated KD of IBTK in multiple cancer cell lines did not alter the levels of eIF4A1/2/3 (*Figure 2B and C*). Furthermore, the depletion of IBTK did not result in any changes to the protein half-life of eIF4A1/2/3 (*Figure 2—figure supplement 1D, E*). Surprisingly, the protein levels of PDCD4, a reported proteolytic substrate of the CRL3[IBTK] complex, remained unchanged in IBTK-KO or -KD cells (*Figure 2B and C*).

We observed that IBTK-WT co-expression effectively ubiquitinated eIF4A1 and eIF4A2, while the ΔBTB mutant did not exhibit such effect (*Figure 2D*). This mutant lacks the BTB1 and BTB2 domains (deletion of aa 554–871), which have been previously demonstrated to be essential for binding to CUL3 (*Pisano et al., 2015*). Consistently, endogenous eIF4A1/2 ubiquitination was reduced upon IBTK KO (*Figure 2E*), and we further demonstrated that the CRL3[IBTK] complex catalyzed eIF4A1 ubiquitination in vitro (*Figure 2F*). Intriguingly, neither overexpression nor IBTK KO had any impact on eIF4A3 ubiquitination (*Figure 2D and E*), despite a strong interaction between these two proteins (*Figure 1J and K*). It is worth noting that eIF4A1 and eIF4A2 have highly similar protein sequences (>90% identity), but eIF4A1 tends to be more abundant than eIF4A2 in most tissues. Furthermore, eIF4A2 KO did not affect cell viability or global protein synthesis (*Galicia-Vázquez et al., 2012*; *Galicia-Vázquez et al., 2015*). Therefore, this study primarily focused on investigating the functional consequences of CRL3[IBTK]-mediated eIF4A1 ubiquitination.

After identifying that CRL3[IBTK]-mediated eIF4A1 ubiquitination is non-degradative, we explored the specificity of polyubiquitin chain linkage on eIF4A1 catalyzed by CRL3[IBTK]. To do this, we conducted ubiquitination assays using a panel of ubiquitin mutants with a single lysine-to-arginine (KR) mutation at seven lysine residues or only one lysine present with the other six lysine residues mutated to arginine (KO). Additionally, we included a ubiquitin mutant in which all lysine residues were replaced with arginine (K-ALL-R). Strikingly, co-expression of any Ub-KR or -KO mutants did not significantly impact IBTK-mediated eIF4A1 ubiquitination, whereas co-expression of the Ub K-ALL-R mutant, which is unable to form polyubiquitin chains, caused only a moderate reduction in IBTK-mediated eIF4A1 ubiquitination (*Figure 2G*). The C-terminal glycine-glycine (GG) residues are essential for Ub conjugation to targeted proteins (*Hershko and Ciechanover, 1998*; *Komander and Rape, 2012*). Indeed, co-expression of the Ub-ΔGG mutant totally abolished IBTK-mediated eIF4A1 ubiquitination (*Figure 2G*). At last, we aimed to identify the specific lysine sites on eIF4A1 where ubiquitin was attached. Using MS to analyze the immunoprecipitated eIF4A1-Ub conjugates, we observed that eIF4A1 was ubiquitinated at 12 lysine residues (*Figure 2—figure supplement 1F*, *Supplementary file 1c*). By introducing mutations that replaced all of these lysine residues to arginine, we observed that the IBTK-mediated ubiquitination of eIF4A1 was completely abolished (*Figure 2—figure supplement 1G*). This suggests

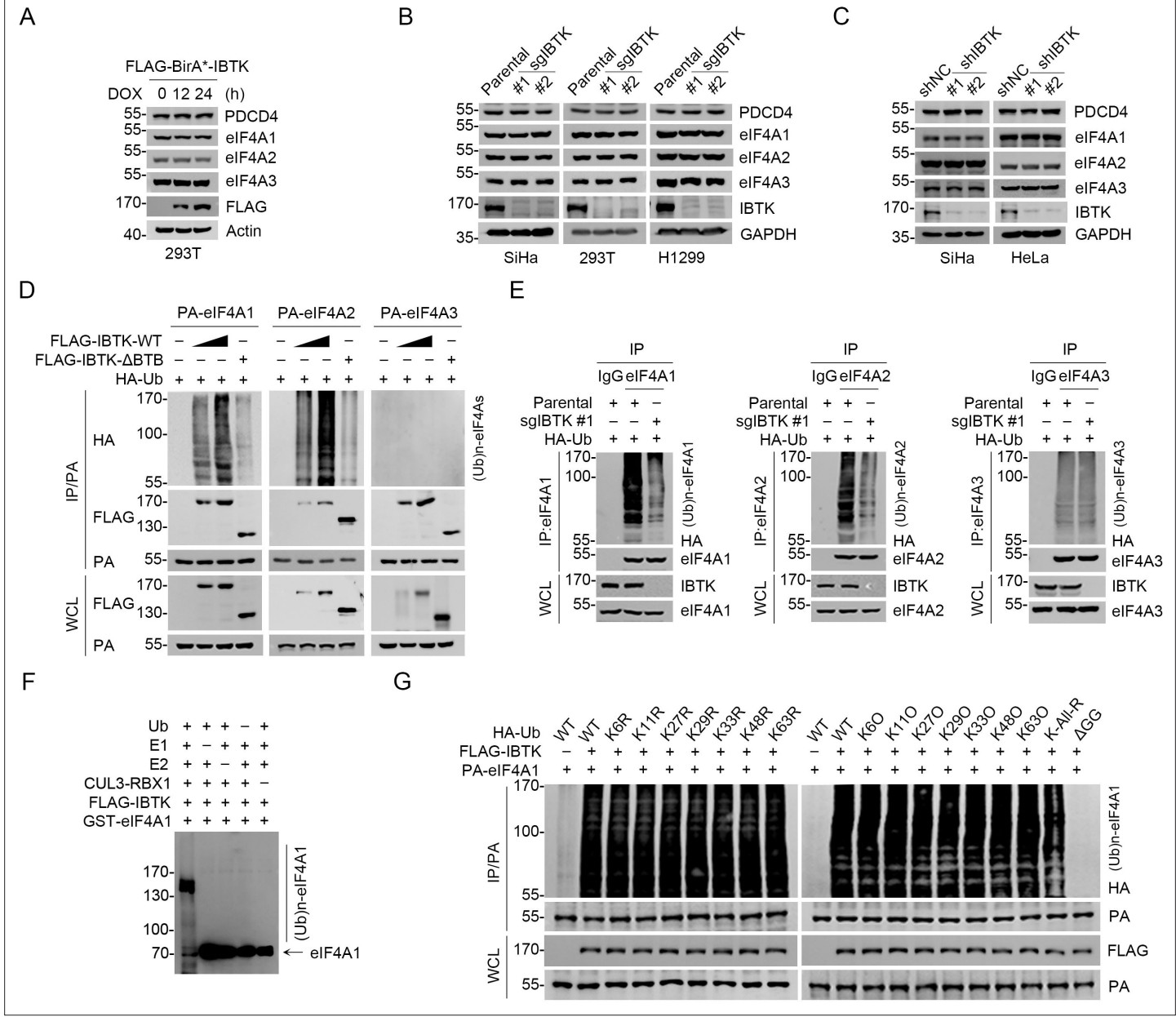

**Figure 2.** Inhibitor of Bruton's tyrosine kinase (IBTK) promotes non-degradative ubiquitination of eIF4A1. (**A**) Western blot (WB) analysis of the indicated proteins in the whole cell lysates (WCLs) from FLAG-BirA*-IBTK Tet-on-inducible T-REx 293 cells treated with doxycycline (DOX) (10 ng/ml) for the indicated times. (**B**) WB analysis of the indicated proteins in the WCLs from parental/IBTK-KO SiHa, 293T, and H1299 cells. (**C**) WB analysis of the indicated proteins in the WCLs from SiHa and HeLa cells infected with lentivirus expressing IBTK-specific shRNA (#1, #2) or negative control (NC). (**D**) WB analysis of the products of in vivo ubiquitination assays from 293T cells transfected with the indicated plasmids. (**E**) Parental or IBTK-KO 293T cells were transfected with HA-Ub for 24 hr, then the WCLs were prepared for co-immunoprecipitation (co-IP) with IgG, anti-eIF4A1, eIF4A2, or eIF4A3 antibody. The polyubiquitinated forms of eIF4A1/2/3 were detected by WB with anti-HA antibody. (**F**) WB analysis of the products of in vitro ubiquitination assays with anti-eIF4A1 antibody. (**G**) WB analysis of the products of in vivo ubiquitination assays from 293T cells transfected with the indicated plasmids.

The online version of this article includes the following source data and figure supplement(s) for figure 2:

**Source data 1.** Original file for the western blot analysis in *Figure 2*.

**Source data 2.** Labeled file for the western blot analysis in *Figure 2*.

**Figure supplement 1.** Inhibitor of Bruton's tyrosine kinase (IBTK) promotes non-degradative ubiquitination of eIF4A1.

**Figure supplement 1—source data 1.** Original file for the western blot analysis in *Figure 2—figure supplement 1*.

**Figure supplement 1—source data 2.** Labeled file for the western blot analysis in *Figure 2—figure supplement 1*.

that some, if not all, of these lysine residues may function as target sites for ubiquitin attachment by the CRL3[IBTK] complex.

Collectively, these data indicate that the CRL3[IBTK] complex mainly catalyzes multi-mono-ubiquitination on eIF4A1, which does not lead to proteasomal degradation.

## IBTK promotes nascent protein synthesis and cap-dependent translation initiation

Given the significance of eIF4A1 in translational regulation, we conducted puromycin incorporation assays to assess the impact of IBTK depletion on global protein synthesis. We observed a noticeable decrease in nascent protein synthesis in IBTK-KO or -KD cells (*Figure 3A and B*). However, this effect was reversed by reintroducing IBTK into IBTK-KO cells (*Figure 3—figure supplement 1A*). Remarkably, the eIF4A1 inhibitor silvestrol also led to a considerable reduction in nascent protein synthesis, particularly in IBTK-KD or KO cells (*Figure 3A and B*). Silvestrol treatment still decreased nascent protein synthesis in IBTK-KO cells overexpressing FLAG-IBTK as well (*Figure 3—figure supplement 1B*). Although eIF4A1 appears to function as a major downstream target of IBTK, overexpressing eIF4A1 was unable to overcome the reduction in puromycin incorporation caused by IBTK deficiency (*Figure 3—figure supplement 1C*).

A decline in translation initiation is often associated with the formation of cytoplasmic stress granules (SGs), which are aggregates resulting from the phase separation of stalled mRNAs and related factors from the surrounding cytosol (*Mahboubi and Stochaj, 2017*). We treated cells with sodium arsenite (AS), a commonly used SG inducer that produces oxidative stress, and examined SG assembly using eIF4A1 or Caprin1 staining as a surrogate. We observed that IBTK-KO cells exhibited significantly higher levels of AS-induced SGs compared to parental cells (*Figure 3C and D*). Conversely, overexpression of IBTK-WT resulted in a marked reduction in AS-induced SG assembly, while the ΔBTB mutant had no such effect (*Figure 3E and F*, *Figure 3—figure supplement 1D*). In contrast, IBTK overexpression had no influence on P-body assembly, as assessed by EDC4 staining (*Figure 3—figure supplement 1E and F*).

We aimed to determine whether IBTK plays a regulatory role in the cap-dependent translation initiation. In mammalian cells, two mechanisms of translation initiation exist: 5' cap-dependent translation initiation and internal ribosomal entry site (IRES)-mediated initiation. To evaluate the effect of IBTK on both mechanisms, we performed dual-luciferase assays. The upstream reporter (Renilla luciferase) of the bicistronic mRNA was translated via a cap-dependent mechanism, whereas the downstream reporter (Firefly luciferase) was under the control of internal initiation from the hepatitis C virus-IRES or (EMCV)-IRES (*Sonenberg, 1990*). The dual-luciferase assay results showed that IBTK depletion caused a significant decrease of Renilla signals by approximately 60% in both systems (*Figure 3G*), indicating that IBTK has a direct effect on cap-dependent translation initiation. Additionally, we noted that IBTK deficiency diminished the interactions between eIF4A1 and other eIF4 subunits (*Figure 3H*), as well as the interactions between capped mRNAs and the eIF4 complex, as indicated by m$^7$GTP-Sepharose pull-down assays (*Figure 3I*).

Collectively, these data indicate that IBTK plays a positive role in regulating the translation initiation activity of the eIF4F complex.

## IBTK deficiency reduces the expression of oncogenes that are dependent on eIF4A1 and mitigates neoplastic phenotypes in cancer cells

The mRNAs of many oncogenes have long and complex 5'-UTRs, which require high levels of eIF4A1 helicase activity for efficient translation (*Boussemart et al., 2014*). In SiHa cells, IBTK KO resulted in a marked reduction in the protein levels of multiple eIF4A1-regulated oncogenes, such as CDK6, MYC, STAT1, and CCND3 (*Figure 4A*). Similar results were observed in IBTK-KO H1299 cells and IBTK-KD CT26 cells (murine colon adenocarcinoma) (*Figure 4—figure supplement 1A–D*). Importantly, the mRNA levels of these eIF4A1-regulated oncogenes were largely unaffected by IBTK deficiency (*Figure 4B*), implying that IBTK exerts a post-transcriptional regulation. Additionally, IBTK overexpression only marginally affected the protein levels of eIF4A1-regulated oncogenes (*Figure 4C*), likely as a result of the consistently high activity of the eIF4 complex in immortalized cell lines. We also showed

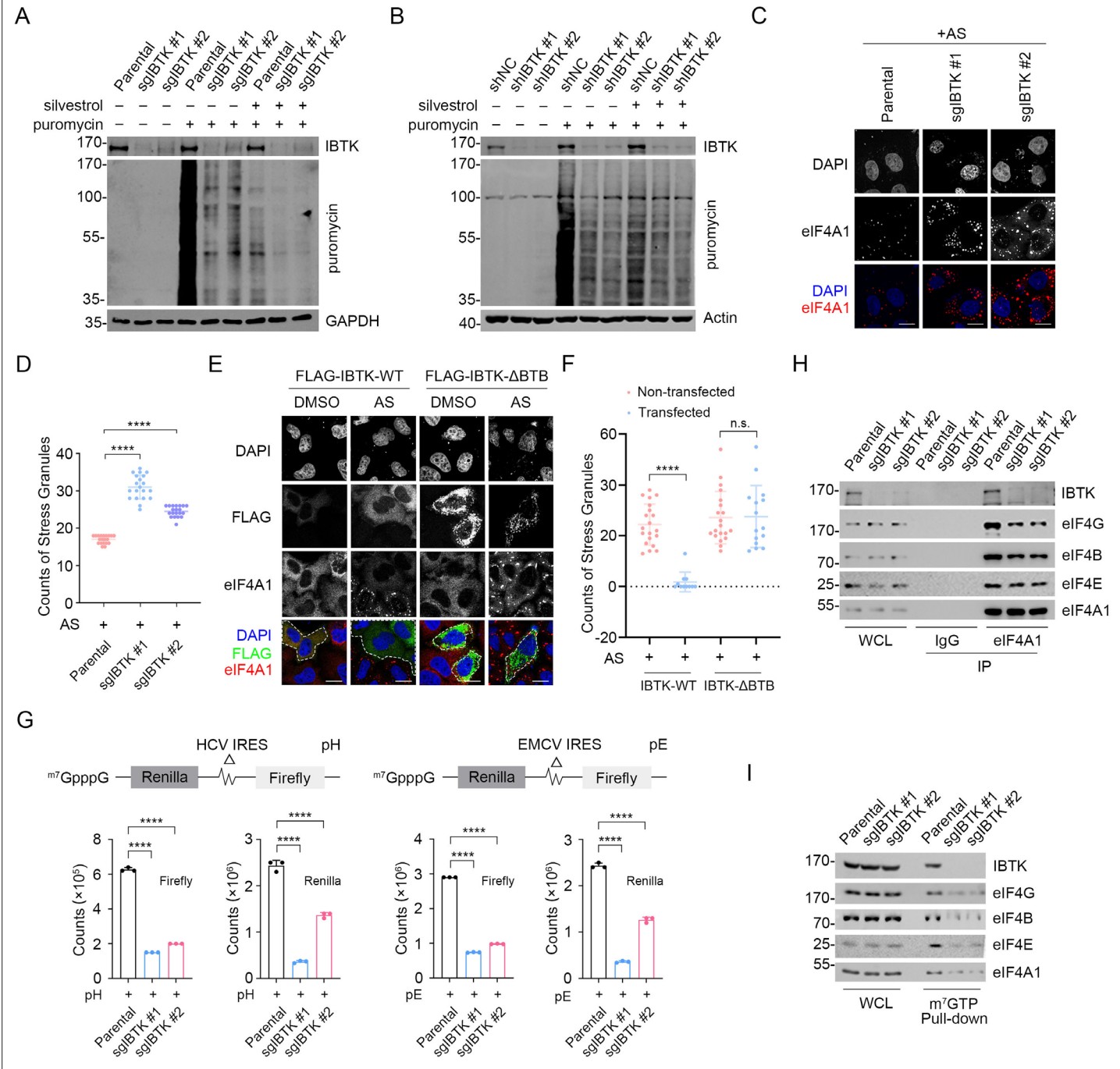

**Figure 3.** Inhibitor of Bruton's tyrosine kinase (IBTK) promotes nascent protein synthesis and cap-dependent translation initiation. (**A**) Western blot (WB) analysis of the indicated proteins in the whole cell lysates (WCLs) from parental and IBTK-KO SiHa cells treated with silvestrol (100 nM, 24 hr) and puromycin (1.5 µM, 10 min). (**B**) WB analysis of the indicated proteins in the WCLs from parental and IBTK-KD HeLa cells treated with silvestrol (100 nM, 24 hr) and puromycin (1.5 µM, 10 min). (**C**) Representative IF images of parental and IBTK-KO SiHa cells treated with arsenite (AS) (100 µM, 2 hr) and then stained with anti-eIF4A1 (red) and DAPI (blue). Scale bar, 20 µm. (**D**) The number of eIF4A1 puncta per cell was used to quantify stress granules in (**C**). Data are presented as means ± SD (n≥10). (**E**) Representative IF images of HeLa cells transfected with the indicated plasmids, treated with DMSO or AS (100 µM, 2 hr), and then stained with anti-FLAG (green), anti-eIF4A1 (red), and DAPI (blue). The cells successfully transfected with FLAG-IBTK are marked with white dashed lines. Scale bar, 20 µm. (**F**) Stress granules (SGs) in (**E**) were quantified by count of eIF4A1 puncta per cell (IBTK transfected vs non-transfected). Data are presented as means ± SD (n≥10). (**G**) Parental or IBTK-KO SiHa cells were transfected with the reporter pRΔDE·HCVF (pH) or pRΔDE·EMCVF (pE) for 24 hr and the luciferase activities were measured. The counts of Firefly and Renilla were measured and shown in the graph. Data are presented as means ± SD (n=3). (**H**) Co-immunoprecipitation (Co-IP) assays using IgG or anti-eIF4A1 antibody in the WCLs prepared from parental and IBTK-KO SiHa cells followed by WB analysis with the indicated antibodies. (**I**) The WCLs of parental and IBTK-KO SiHa cells were incubated with

*Figure 3 continued on next page*

*Figure 3 continued*

m$^7$GTP-Sepharose beads, and the pull-downed proteins were subjected to WB analysis with the indicated antibodies. p Values are calculated using one-way analysis of variance (ANOVA) test in (**D, F, G**). ****p<0.0001, n.s. non-significant.

The online version of this article includes the following source data and figure supplement(s) for figure 3:

**Source data 1.** Original file for the western blot analysis in *Figure 3*.

**Source data 2.** Labeled file for the western blot analysis in *Figure 3*.

**Figure supplement 1.** Inhibitor of Bruton's tyrosine kinase (IBTK) promotes nascent protein synthesis and cap-dependent translation initiation.

**Figure supplement 1—source data 1.** Original file for the western blot analysis in *Figure 3—figure supplement 1*.

**Figure supplement 1—source data 2.** Labeled file for the western blot analysis in *Figure 3—figure supplement 1*.

that the impacts of IBTK deficiency on reducing protein expression of eIF4A1 targets were unable to be reversed by eIF4A1 overexpression (*Figure 4—figure supplement 1E*).

Given the crucial role of eIF4A1 in cancer biology, we conducted additional investigations to determine if IBTK promotes neoplastic phenotypes in cancer cells. We observed that IBTK deficiency significantly decreased SiHa cell proliferation, migration, invasion, and anchorage-independent cell growth (*Figure 4D–G*). IBTK depletion reduced xenograft tumor growth in vivo (*Figure 4H*). Moreover, IBTK deficiency enhanced apoptosis in response to eIF4A1 inhibitors (silvestrol and rocaglamide A) (*Figure 4I and J*, *Figure 4—figure supplement 2A and B*). The tumor-suppressive effects of IBTK depletion in HeLa cells were also observed (*Figure 4—figure supplement 2C–H*).

Collectively, these data indicate that IBTK plays critical roles in promoting oncogene expression and cancer cell malignancy by upregulating eIF4A1 activity.

## IBTK facilitates IFN-γ-induced PD-L1 expression and tumor immune escape

A control mechanism of tumor immune evasion was recently demonstrated at the translational level through the eIF4F-STAT1-IRF1-PD-L1 pathway. The eIF4F complex regulates the translation of transcription factor STAT1 and promotes IFN-γ-induced PD-L1 expression (*Cerezo et al., 2018*). As IBTK-KO cells showed compromised eIF4F-initiated translation activity and downregulated STAT1 protein levels, we investigated the potential impacts of IBTK on IFN-γ-induced PD-L1 expression and tumor immune escape. We found that the IFN-γ-inducible expression of STAT1, IRF1, and PD-L1 was markedly reduced in both IBTK-KO H1299 cells (*Figure 4—figure supplement 3A–C*) and IBTK-KD CT26 cells (*Figure 4—figure supplement 3D–F*). To further investigate the effects of IBTK in tumor growth in the presence of a functional immune system, we utilized a CT26 xenograft tumor model and found that immunocompetent mice implanted with IBTK-KD CT26 cells exhibited reduced tumor growth in vivo (*Figure 4—figure supplement 3G–I*). CD8[+] cytotoxic T lymphocytes (CTLs) play a crucial role in antitumor immunity by secreting granzyme B, which triggers apoptosis. Notably, depletion of IBTK increased the population of CD8[+] CTLs and the release of granzyme B in xenograft tumors (*Figure 4—figure supplement 3J and K*).

Collectively, these data indicate that IBTK promotes IFN-γ-inducible PD-L1 expression and tumor immune escape.

## IBTK-mediated eIF4A1 ubiquitination is regulated by the mTORC1/S6K1 signaling

We investigated whether the upstream mTORC1/S6K1 signaling has any impact on IBTK-mediated eIF4A1 ubiquitination. Three quantitative phosphoproteomic studies revealed that treatment with mTOR inhibitors (rapamycin or Torin 1) resulted in a significant reduction in the phosphorylation levels at seven Ser/Thr sites on IBTK (*Chen et al., 2009*; *Hsu et al., 2011*; *Yu et al., 2011*). Intriguingly, these sites cluster within a narrow region (990–1068 aa) proximate to the second BTB domain of IBTK (*Figure 5A*). We confirmed the interactions of IBTK with the mTORC1 complex and S6K1 (*Figure 5B–E*). Subsequently, we examined the impact of mTORC1 signaling on IBTK phosphorylation. For this purpose, we generated an IBTK deletion mutant (900–1150 aa) encompassing the putative phosphorylation sites regulated by mTOR. Phos-tag gels were employed to detect changes in the electrophoretic mobility of phosphorylated proteins. The findings indicated a marked decrease

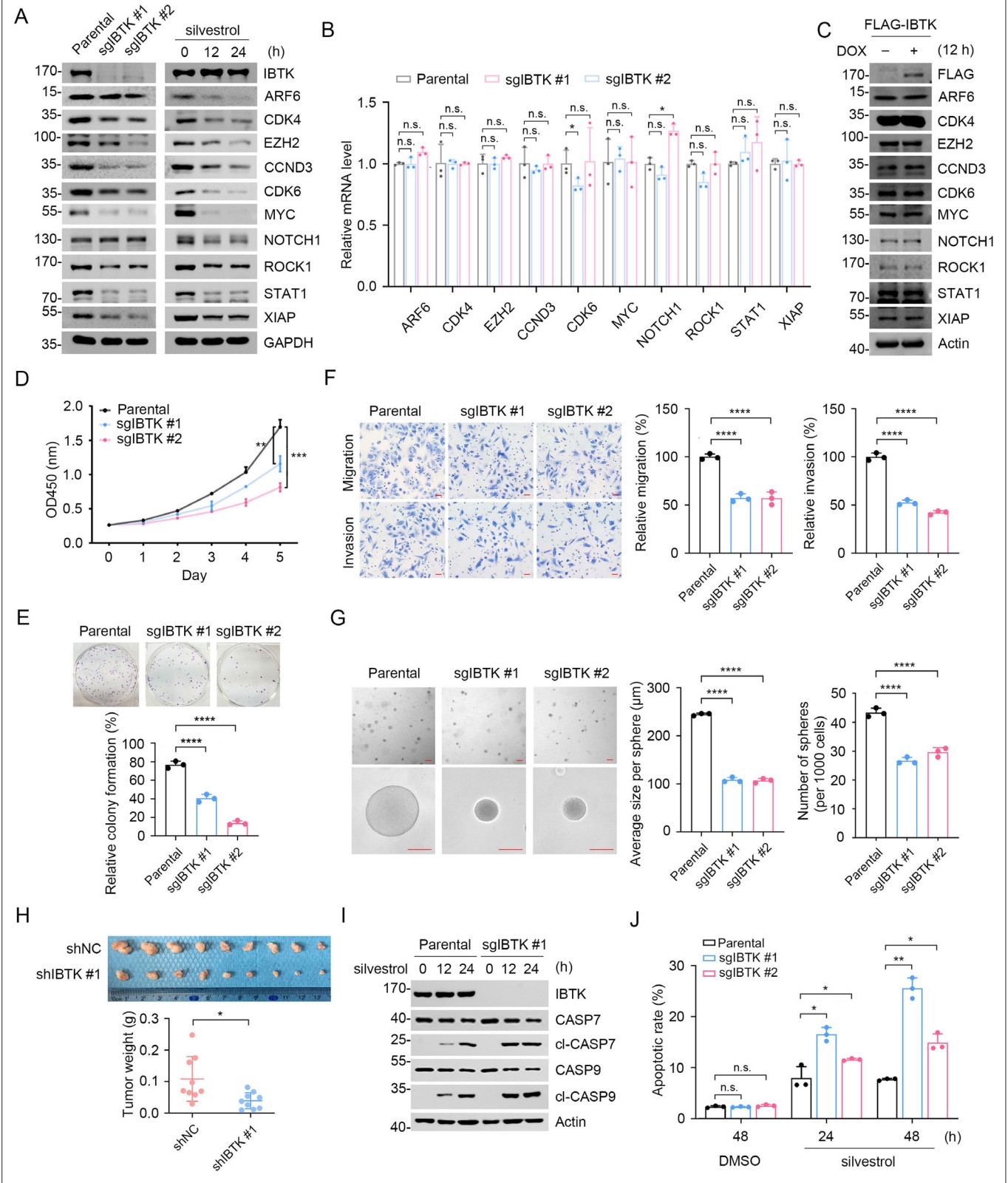

**Figure 4.** Inhibitor of Bruton's tyrosine kinase (IBTK) deficiency reduces eIF4A1-dependent oncoprotein expression and neoplastic phenotypes in cancer cells. (**A**) Western blot (WB) analysis of the indicated proteins in the whole cell lysates (WCLs) from parental and IBTK-KO SiHa cells (left panel) and the WCLs from SiHa cells treated with silvestrol (100 nM) for the indicated times (right panel). (**B**) Quantitative reverse transcription polymerase chain reaction (RT-qPCR) assessment of the mRNA expression of eIF4A1 target genes in parental and IBTK-KO SiHa cells. The mRNA levels of *Actin*

*Figure 4 continued on next page*

*Figure 4 continued*

were used for normalization. Data are shown as means ± SD (n=3). (**C**) WB analysis of the indicated proteins in the WCLs from FLAG-BirA*-IBTK Tet-on-inducible Flp-In T-REx 293 cells treated with DOX (10 ng/ml) for 12 hr. (**D**) Cell Counting Kit-8 (CCK-8) cell proliferation analysis of parental and IBTK-KO SiHa cells. Data are shown as means ± SD (n=3). (**E**) Colony formation analysis of parental and IBTK-KO SiHa cells, and the quantitative data is shown (below). Data are shown as means ± SD (n=3). (**F**) Cell migration and invasion analysis of parental and IBTK-KO SiHa cells, and the quantitative data is shown (right). Data are shown as means ± SD (n=3). Scale bar, 50 μm. (**G**) Sphere-formation analysis of parental and IBTK-KO SiHa cells. Representative pictures of SiHa cells after 2 weeks in three-dimensional culture are shown. Average size per sphere and number of spheres per 1000 cells were calculated by ImageJ. The quantitative data is shown (right). Data are shown as means ± SD (n=3). Scale bar, 100 μm. (**H**) Parental and IBTK-KD HeLa cells were injected subcutaneously (s.c.) into the right flank of BALB/c mice. The tumors in each group at day 20 were harvested and photographed, the quantitative data of tumor weights is shown (below). Data are shown as means ± SD (n=9). (**I**) WB analysis of the indicated proteins in the WCLs from parental and IBTK-KO SiHa cells treated with silvestrol (100 nM) for the indicated times. (**J**) Parental and IBTK-KO SiHa cells were treated with silvestrol (100 nM) for the indicated times. Then, annexin-V-FITC/PI dyes were used to stain the harvested cells, after which flow cytometry analysis was performed. Data are shown as means ± SD (n=3). p Values are calculated using unpaired Student's t-test in (**H**), one-way analysis of variance (ANOVA) test in (**E, F, G**), two-way ANOVA test in (**B, D, J**). *p<0.05, **p<0.01, ***p<0.001, ****p<0.0001, n.s. non-significant.

The online version of this article includes the following source data and figure supplement(s) for figure 4:

**Source data 1.** Original file for the western blot analysis in *Figure 4*.

**Source data 2.** Labeled file for the western blot analysis in *Figure 4*.

**Figure supplement 1.** Inhibitor of Bruton's tyrosine kinase (IBTK) is indispensable for eIF4A1-related oncogene expression.

**Figure supplement 1—source data 1.** Original file for the western blot analysis in *Figure 4—figure supplement 1*.

**Figure supplement 1—source data 2.** Labeled file for the western blot analysis in *Figure 4—figure supplement 1*.

**Figure supplement 2.** Inhibitor of Bruton's tyrosine kinase (IBTK) deficiency reduces neoplastic phenotypes in cancer cells.

**Figure supplement 2—source data 1.** Original file for the western blot analysis in *Figure 4—figure supplement 2*.

**Figure supplement 2—source data 2.** Labeled file for the western blot analysis in *Figure 4—figure supplement 2*.

**Figure supplement 3.** Inhibitor of Bruton's tyrosine kinase (IBTK) is required for IFN-γ-induced PD-L1 expression.

**Figure supplement 3—source data 1.** Original file for the western blot analysis in *Figure 4—figure supplement 3*.

**Figure supplement 3—source data 2.** Labeled file for the western blot analysis in *Figure 4—figure supplement 3*.

in phosphorylated forms FLAG-IBTK$_{900-1150aa}$, while the non-phosphorylated form exhibited a simultaneous increase in cells deprived of amino acids or treated with rapamycin (*Figure 5F*). We also treated FLAG-IBTK$_{900-1150aa}$ overexpressed cells with lambda phosphatase. As shown in *Figure 5G*, lambda phosphatase treatment completely abolished the mobility shift of phosphorylated FLAG-IBTK$_{900-1150aa}$. Additionally, the lowest band displayed an abundant accumulation of the non-phosphorylated form of FLAG-IBTK$_{900-1150aa}$. These findings confirmed that the mobility shift observed in WB analysis corresponds to phosphorylated forms of this deletion mutant. Moreover, the phosphorylation levels of IBTK$_{900-1150aa}$ were markedly downregulated in cells deficient in Raptor, an essential subunit of mTORC1 but not mTORC2 (*Figure 5H*), indicating that the mTORC1 signaling plays a predominant role in mediating IBTK phosphorylation.

To investigate whether mTOR or S6K1 directly phosphorylates IBTK, we conducted in vitro kinase assays using recombinant IBTK$_{900-1150aa}$ segment as a substrate. The analysis of the phospho-peptides via MS revealed that mTOR phosphorylates residues S911/999/1004/1039/1045/1069/1089/1141, while S6K1 phosphorylates residues S971/990/993/1033/1045/1096 and T1101 (*Figure 5I and J*, *Figure 5—figure supplement 1A and B*, *Supplementary file 1d and e*). Thus, four out of seven sites (excluding S992, T1008, and T1068) were confirmed as mTORC1/S6K1 phosphorylation sites in vitro, thereby confirming that IBTK is an authentic mTORC1/S6K1 substrate. Then, we generated an IBTK mutant (IBTK-7S/TA) in which seven Ser/Thr were replaced with Ala to abolish phosphorylation regulated by mTORC1/S6K1. Simultaneous mutation of seven Ser/Thr sites to Ala nearly abolished the mobility shift of IBTK$_{900-1150aa}$ (*Figure 5K*), indicating that most, if not all, mTOR-regulated sites are localized in this region. Recognizing that S990/992/993 are three adjacent sites, we deemed it appropriate to generate a single antibody to recognize the phospho-S990/992/993 epitope. Using this antibody, we observed a marked decrease in the phosphorylation levels of three adjacent Ser residues in IBTK upon AA deprivation or rapamycin treatment (*Figure 5L*). Furthermore, IBTK-mediated eIF4A1 ubiquitination was considerably decreased upon AA deprivation or rapamycin treatment (*Figure 5M*). This decrease was also evident in Raptor-KO cells (*Figure 5N*). Additionally, the IBTK-7S/TA mutant showed diminished capacity to ubiquitinate eIF4A1 compared to IBTK-WT (*Figure 5O*).

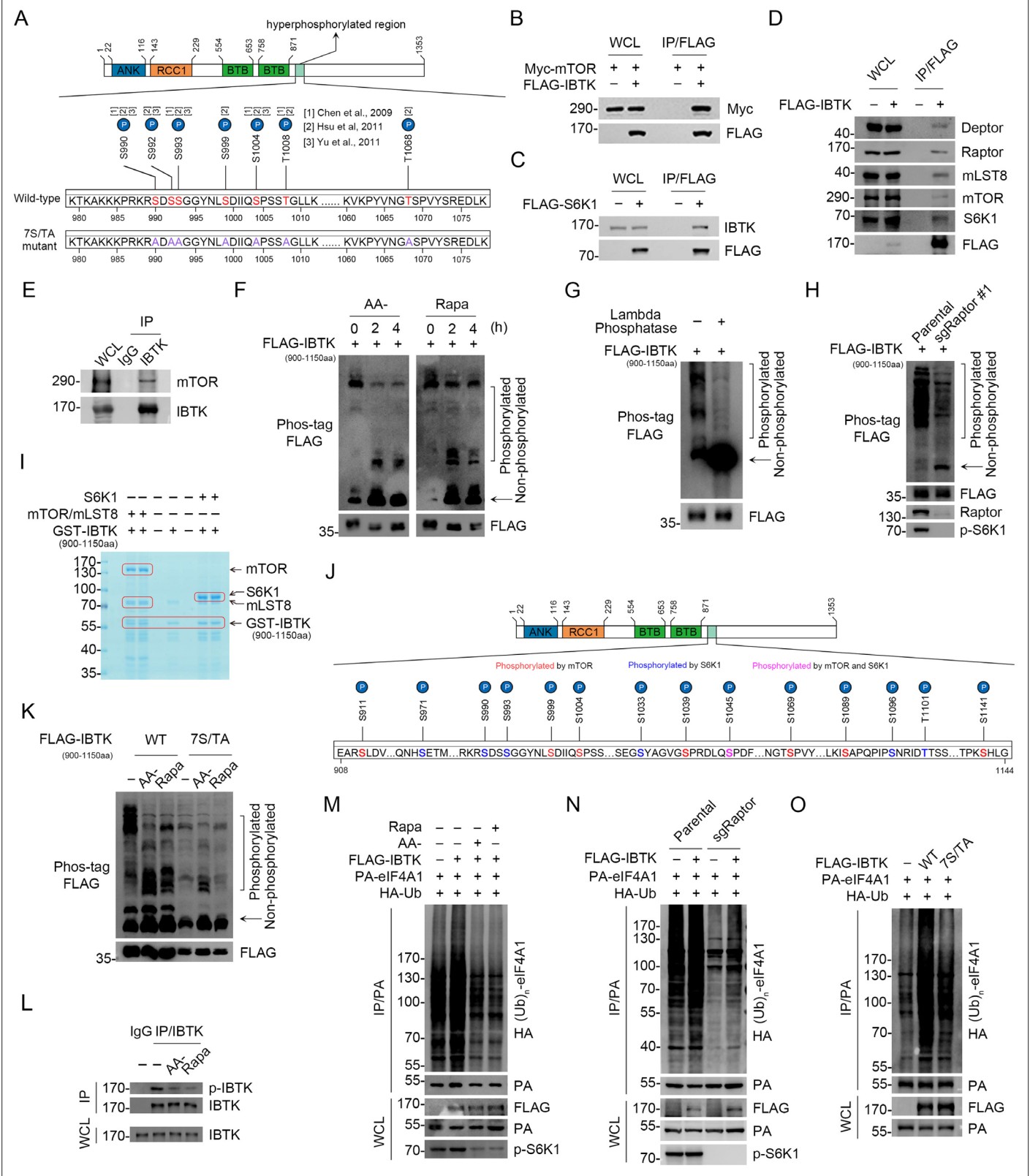

**Figure 5.** Inhibitor of Bruton's tyrosine kinase (IBTK)-mediated eIF4A1 ubiquitination is regulated by the mTORC1/S6K1 signaling. (**A**) The potential mTOR-regulated phosphorylation sites of IBTK which were sensitive to rapamycin or Torin1 treatment, according to three quantitative phosphoproteomic studies. (**B–D**) Western blot (WB) analysis of the indicated proteins in the whole cell lysates (WCLs) and co-immunoprecipitation (co-IP) samples of anti-FLAG antibody obtained from 293T cells transfected with the indicated plasmids. (**E**) Co-IP using anti-IBTK antibody in the WCLs

*Figure 5 continued on next page*

*Figure 5 continued*

prepared from 293T cells, followed by WB analysis with the indicated antibodies. (**F**) WB analysis of the indicated proteins in the WCLs from 293T cells transfected with FLAG-IBTK$_{900-1150aa}$ and starved of amino acids (AA) or treated with rapamycin (500 nM) for the indicated times. The phosphorylated forms of IBTK$_{900-1150aa}$ were detected by WB using phos-tag gels. The arrow indicates non-phosphorylated IBTK$_{900-1150aa}$. (**G**) WB analysis of the indicated proteins in the WCLs from 293T cells treated with lambda phosphatase (70 U/µl) at 30°C for 1 hr. The phosphorylated form of IBTK$_{900-1150aa}$ was detected by WB using phos-tag gels. (**H**) WB analysis of the indicated proteins in the WCLs from parental or Raptor-KO 293T cells transfected with FLAG-IBTK$_{900-1150aa}$. The phosphorylated form of IBTK$_{900-1150aa}$ was detected by WB using phos-tag gels. (**I**) Recombinant GST-IBTK$_{900-1150aa}$ proteins were subjected to phosphorylation by recombinant mTOR/mLST8, or S6K1, as detected using in vitro kinase assays. The reaction products were separated by SDS-PAGE and visualized by Coomassie Blue (CB) staining. The catalytic domain of human mTOR (1362–2549) was used in this assay. (**J**) mTOR/mLST8 or S6K1-mediated IBTK in vitro phosphorylation sites identified by mass spectrometry (MS) analysis. (**K**) WB analysis of the indicated proteins in the WCLs from 293T cells transfected with FLAG-IBTK$_{900-1150aa}$ (WT or 7S/TA mutant) and starved of amino acids or treated with rapamycin (500 nM) for 2 hr. (**L**) Co-IP using anti-IBTK antibody in the WCLs prepared from 293T cells starved of amino acids (AA-) or treated with rapamycin (500 nM) for 2 hr, followed by WB analysis with the indicated antibodies. (**M**) WB analysis of the products of in vivo ubiquitination assays from 293T cells transfected with the indicated plasmids and starved of amino acids or treated with rapamycin (500 nM) for 2 hr. (**N**) WB analysis of the products of in vivo ubiquitination assays from parental or Raptor-KO 293T cells transfected with the indicated plasmids. (**O**) WB analysis of the products of in vivo ubiquitination assays from 293T cells transfected with the indicated plasmids.

The online version of this article includes the following source data and figure supplement(s) for figure 5:

**Source data 1.** Original file for the western blot analysis in *Figure 5*.

**Source data 2.** Labeled file for the western blot analysis in *Figure 5*.

**Figure supplement 1.** mTOR and S6K1 phosphorylates inhibitor of Bruton's tyrosine kinase (IBTK) in vitro.

Collectively, these data indicate that the mTORC1/S6K1 signaling engages in IBTK-mediated eIF4A1 ubiquitination by directly phosphorylating IBTK.

## mTORC1/S6K1-mediated IBTK phosphorylation is crucial for sustaining oncogenic translation and cancer cell malignancy

To elucidate the biological significance of IBTK phosphorylation mediated by the mTORC1/S6K1 pathway, we reintroduced IBTK-WT or IBTK-7S/TA mutant into IBTK-KO SiHa cells (termed IBTK[WT] and IBTK[7S/TA], respectively). We demonstrated that IBTK[7S/TA] cells exhibited lower levels of global protein synthesis and reduced expression of eIF4A1-regulated oncogenes compared to IBTK[WT] cells (*Figure 6A and B*). Additionally, there was a noticeable reduction in the interactions between eIF4A1 and other eIF4 complex subunits in IBTK[7S/TA] cells compared to IBTK[WT] cells (*Figure 6C*). The interactions between capped mRNAs and eIF4 complex subunits were also markedly reduced in IBTK[7S/TA] cells (*Figure 6D*). Furthermore, IBTK[7S/TA] cells displayed a substantial decrease in cell growth and migration (*Figure 6E–I*), while showing an increase in silvestrol-induced apoptosis compared to IBTK[WT] cells (*Figure 6J*).

Collectively, these data indicate that mTORC1/S6K1-mediated IBTK phosphorylation favors sustained oncogenic translation and cancer cell malignancy.

## Overexpression of IBTK correlates with poor survival in cervical cancer

The aforementioned results established that IBTK promotes oncogenic eIF4A1 activation in cervical cancer cell lines (HeLa and SiHa). In light of these findings, we sought to evaluate the pathological significance of IBTK expression in cervical cancer. The Cancer Genome Atlas (TCGA) dataset revealed that higher IBTK mRNA expression levels were correlated with poor survival in cervical squamous cell carcinoma (CESC), which is the most prevalent histological subtype of cervical cancer (*Figure 7A*). Following validation of the antibody specificity for immunohistochemistry (IHC) analysis in parental/IBTK-KO cells (*Figure 7—figure supplement 1*), we conducted an IHC staining analysis of 117 CESC tissues and 35 adjacent normal cervical tissues, by using a tissue microarray (*Figure 7B*). The results showed a marked upregulation of IBTK protein expression in CESC tissues compared to adjacent normal cervical tissues (*Figure 7C*). Furthermore, higher levels of IBTK expression were positively correlated with advanced pathological grades and were indicative of poorer survival outcomes (*Figure 7D and E*).

Collectively, these data indicate that elevated IBTK expression is evident in CESC, correlating with a poor prognosis for patients with CESC.

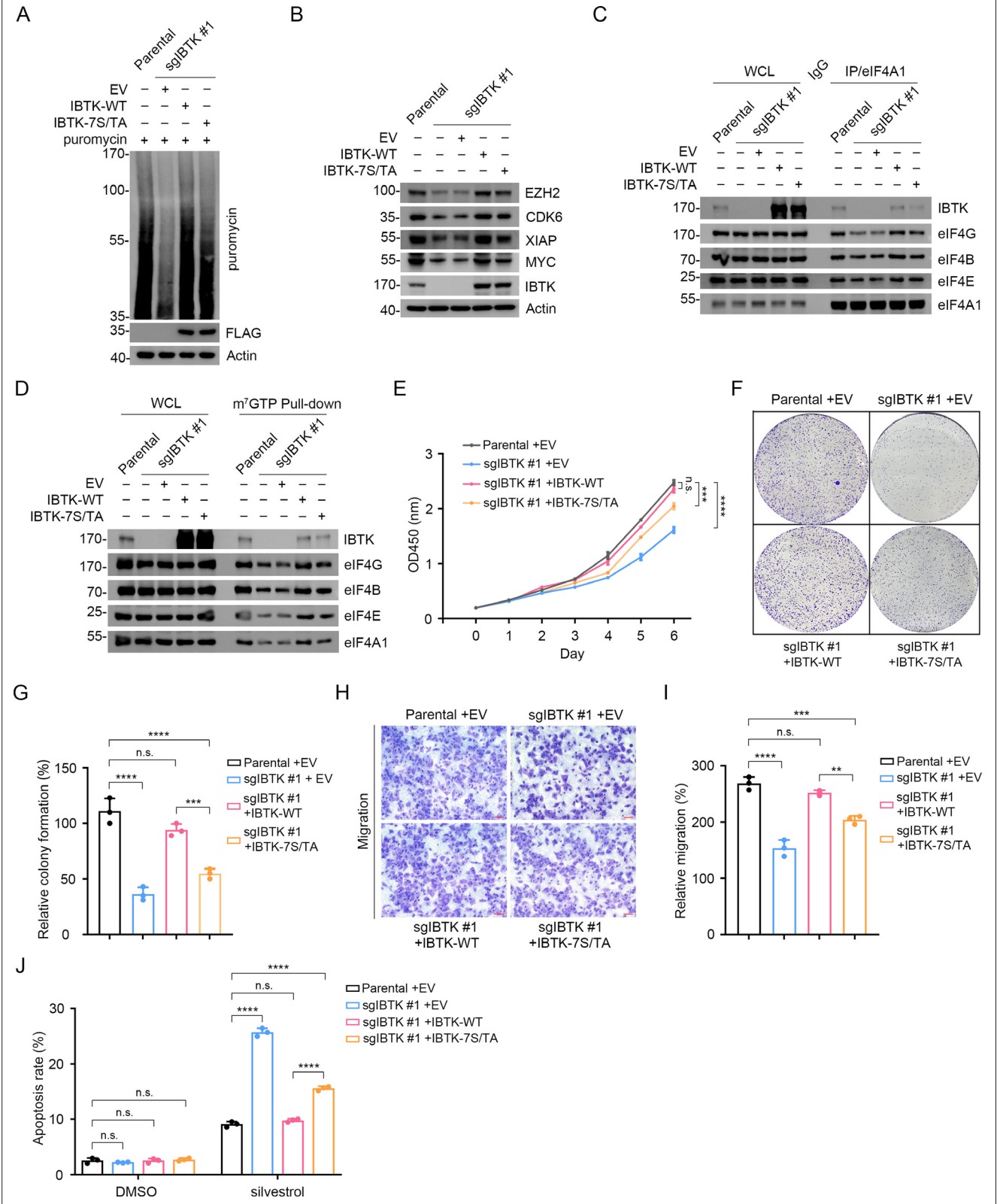

**Figure 6.** Deficiency in mTORC1/S6K1-mediated eIF4A1 phosphorylation reduces eIF4A1-dependent oncoprotein expression and neoplastic phenotypes in cancer cells. (**A**) Western blot (WB) analysis of the indicated proteins in the whole cell lysates (WCLs) from parental and IBTK-KO SiHa cells reconstituted with EV, IBTK-WT, or -7S/TA mutant and treated with puromycin (1.5 µM, 10 min). (**B**) WB analysis of the indicated proteins in the WCLs from parental and IBTK-KO SiHa cells reconstituted with EV, IBTK-WT, or -7S/TA mutant. (**C**) Co-immunoprecipitation (Co-IP) assays using anti-

*Figure 6 continued on next page*

*Figure 6 continued*

eIF4A1 antibody or IgG in the WCLs prepared from parental and IBTK-KO SiHa cells followed by WB analysis with the indicated antibodies. (**D**) The WCLs of parental and IBTK-KO SiHa cells reconstituted with EV, IBTK-WT, or -7S/TA mutant were incubated with $m^7$GTP-Sepharose beads, and the pull-down proteins were subjected to WB analysis with the indicated antibodies. (**E**) Cell Counting Kit-8 (CCK-8) cell proliferation analysis of parental and IBTK-KO SiHa cells reconstituted with EV, IBTK-WT, or -7S/TA mutant. Data are shown as means ± SD (n=3). (**F, G**) Colony formation analysis of parental and IBTK-KO SiHa cells reconstituted with EV, IBTK-WT or, -7S/TA mutant, and the quantitative data is shown in (**G**). Data are shown as means ± SD (n=3). (**H, I**) Cell migration analysis of parental and IBTK-KO SiHa cells reconstituted with EV, IBTK-WT or, -7S/TA mutant, and the quantitative analysis is shown in (**I**). Data are shown as means ± SD (n=3). Scale bar, 100 µm. (**J**) Parental and IBTK-KO SiHa cells reconstituted with EV, IBTK-WT or, -7S/TA mutant were treated with silvestrol (100 nM) for 24 hr. Then, annexin-V-FITC/PI dyes were used to stain the harvested cells, after which flow cytometry analysis was performed. Data are shown as means ± SD (n=3). p Values are calculated using one-way analysis of variance (ANOVA) test in (**G, I**), two-way ANOVA test in (**E, J**). **p<0.01, ***p<0.001, ****p<0.0001, n.s. non-significant.

The online version of this article includes the following source data for figure 6:

**Source data 1.** Original file for the western blot analysis in *Figure 6*.

**Source data 2.** Labeled file for the western blot analysis in *Figure 6*.

## Discussion

Oncogenic signaling appears to dominate translation control at nearly all stages of cancer propagation for very specific and distinct cellular phenotypes. In general, cancer cells upregulate eIF4F activity to promote cap-dependent translation and oncogene expression, leading to rapid cell growth. Our study reveals an uncharacterized signaling axis involving mTORC1/S6K1-IBTK-eIF4A1 that promotes cap-dependent translation and oncoprotein expression, contributing to tumor cell growth. Additionally, we demonstrate that mTORC1 and S6K1 phosphorylate IBTK at multiple sites to boost CRL3$^{IBTK}$-mediated eIF4A1 ubiquitination under nutrient-rich conditions (*Figure 7F*). Overall, our study adds to our understanding of how cancer cells hijack protein synthesis machinery to promote their survival and proliferation, and provides potential avenues for developing new cancer treatments.

The development of eIF4A inhibitors as therapeutic agents for cancer patients is an exciting area of research. However, the emergence of resistance to these inhibitors highlights the need for a better understanding of the molecular mechanisms underlying drug resistance. The recent genome-wide CRISPR/Cas9 screen has identified three negative NRF2 regulators (KEAP1, CUL3, and CAND1) whose inactivation can confer resistance to the silvestrol analogue, providing insights into potential pathways that may contribute to drug resistance (*Sanghvi et al., 2021*). Additionally, our study shows that IBTK overexpression suppresses AS-induced SG assembly and that ablation of IBTK sensitizes cancer cells to eIF4A inhibitor-induced cell death, making IBTK a potential therapeutic target to overcome intrinsic resistance to eIF4A inhibitors. Overall, these findings underscore the importance of continued research in the development of eIF4A inhibitors and the identification of new therapeutic targets to combat drug resistance in cancer cells.

We observed that IBTK depletion indeed led to a substantial reduction in the protein levels of most detected eIF4A1-regulated oncogenes, but there were exceptions. For instance, IBTK KO in H1299 cells exerted minimal influence on the protein levels of ROCK1 (*Figure 4—figure supplement 1A*). Several possible explanations might account for this observation: first, given that our list of eIF4A1 target genes collected from previous studies conducted using distinct cell lines, it is not unexpected for different cell lines to exhibit subtle differences in regulation of eIF4A1 target genes. Second, as a CRL3 adaptor, IBTK potentially possesses other biological functions via ubiquitination of multiple substrates; dysregulation of these could buffer the impact of IBTK deficiency on the protein expression of specific eIF4A1 targets.

The tumor suppressor PDCD4 inhibits translation by binding directly to eIF4A. Pisano et al. reported that PDCD4 is the first CRL3$^{IBTK}$ substrate and that CRL3$^{IBTK}$ targets PDCD4 for ubiquitin-dependent degradation (*Pisano et al., 2015*). Although this study highlighted the potential involvement of IBTK in translation regulation, we did not observe any changes in protein levels of PDCD4 in IBTK-KD or -KO cells. Instead, our study demonstrated that eIF4A1 is a major downstream effector of IBTK, which has been previously observed to regulate several well-characterized eIF4A targets, such as MYC and MCL-1, in Eµ-myc mice (*Vecchio et al., 2019*). Our results also indicate that IBTK may play a role in tumor immune escape through modulating the eIF4F-STAT1-IRF1-PD-L1 axis. This finding is in line with a recent study that eIF4A1 promotes PD-L1-mediated tumor immune escape by facilitating the translation of STAT1 (*Cerezo et al., 2018*).

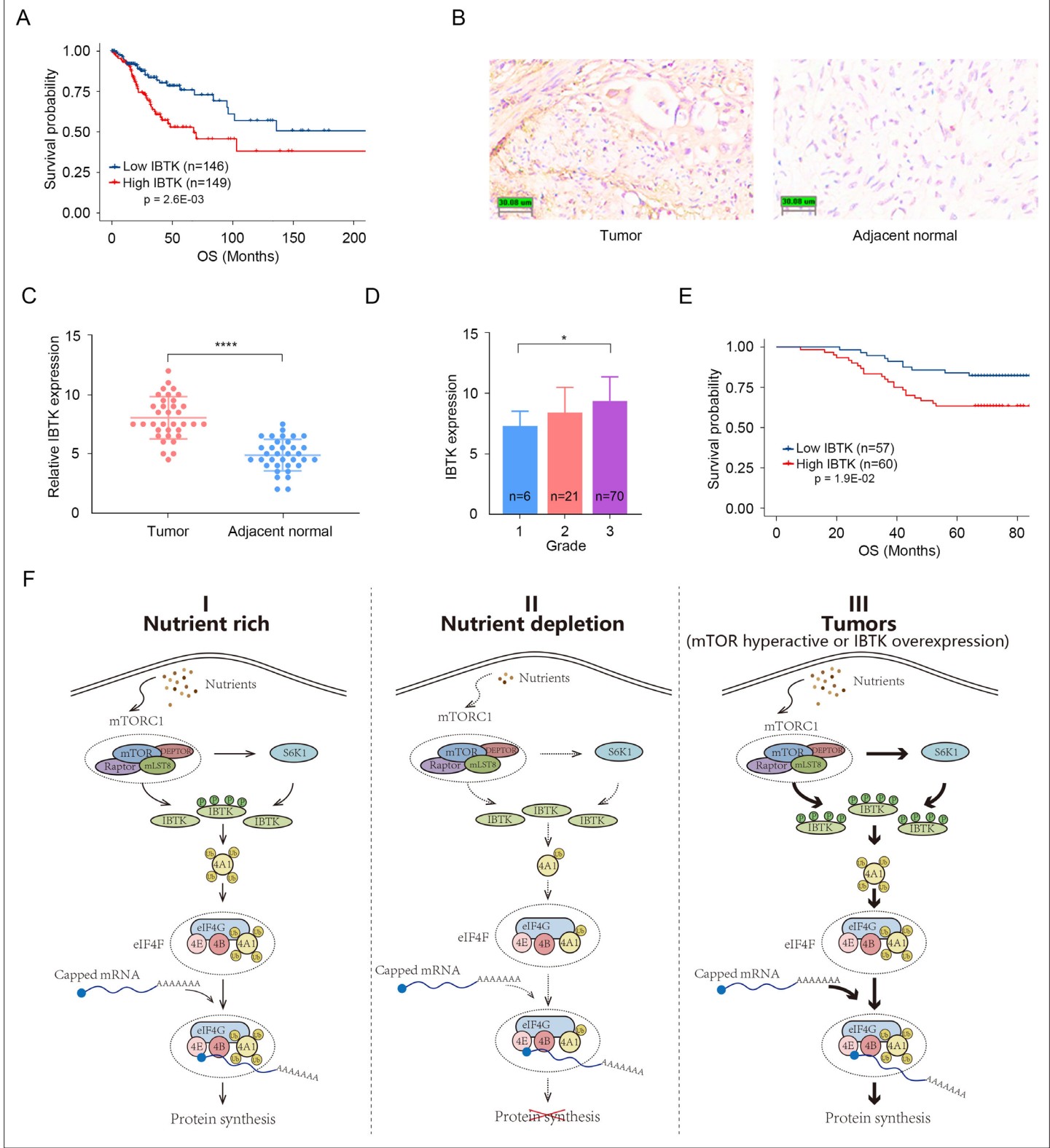

**Figure 7.** Overexpression of inhibitor of Bruton's tyrosine kinase (IBTK) correlates with poor survival in cervical cancer. (**A**) Kaplan-Meier survival plots of overall survival (OS) were analyzed according to IBTK mRNA expression in cervical squamous cell carcinoma (CESC) specimens from The Cancer Genome Atlas (TCGA) cohort (n=295). (**B, C**) Immunohistochemistry (IHC) analysis of IBTK protein expression in CESC tissues (n=35) and adjacent normal cervical tissues (n=35). Representative staining results were shown. Scale bar, 30 μm. The relative intensity of IBTK was measured using ImageJ and the quantitative data is shown in (**C**). (**D**) IBTK expression in samples with different pathological grade. (**E**) Kaplan-Meier survival curves of OS

*Figure 7 continued on next page*

*Figure 7 continued*

comparing high and low expression of IBTK in the TMA (n=117). (**F**) A proposed model depicting that the mTORC1/S6K1 signaling promotes translation initiation by enhancing IBTK-mediated eIF4A1 ubiquitination. p Values in (**A, E**) are calculated using log-rank test while p Values in (**C, D**) are calculated using unpaired Student's t-test. *p<0.05, ****p<0.0001.

The online version of this article includes the following figure supplement(s) for figure 7:

**Figure supplement 1.** Representative images of inhibitor of Bruton's tyrosine kinase (IBTK) immunohistochemistry (IHC) staining in parental and IBTK-KO SiHa cells.

In contrast to eIF4A1/2, IBTK does not possess the ability to ubiquitinate eIF4A3, despite a robust interaction between these two proteins. It remains unclear whether IBTK can facilitate eIF4A3 ubiquitination under specific conditions or merely modulate eIF4A3 activity through direct binding. eIF4A3 is an essential subunit of EJC (*Lu et al., 2014*). The role of IBTK in post-transcriptional gene regulation through modulation of EJC activity is an area that warrants further investigation, particularly given that KD of eIF4A3 or IBTK all had an impact on alternative splicing in HeLa cells (*Fiume et al., 2016*; *Wang et al., 2014*). Alongside its interaction with eIF4A1/2, IBTK may also play diverse roles in different stages of the translation process. The AP-MS analysis and biochemical validation provide strong evidence for the interaction between IBTK and eIF2α (*Figure 1G*). A previous study showed that a functional uORF in the 5'-UTR sequence allows for preferential translation of IBTK in response to p-eIF2α and ER stress (*Amin et al., 2023*). Our findings suggest that IBTK may interact with eIF2α and modulate its activity, either positively or negatively, through direct binding in response to p-eIF2α and ER stress. Furthermore, the identification of IBTK interacting with multiple subunits of eIF3 complex (*Figure 1C and G*), which plays key roles in different stages of translation (*Gomes-Duarte et al., 2018*), highlights the potential role of IBTK in modulating translation through the eIF3 complex. Taken together, these findings provide a molecular basis for comprehensive investigations into the precise roles of IBTK in controlling the efficiency and fidelity of protein synthesis.

Several quantitative phosphoproteomic studies have used mTOR inhibitors coupled with MS to demonstrate that mTOR modulates the phosphorylation of thousands of currently uncharacterized mTOR substrates, either directly or indirectly through its downstream kinases (*Chen et al., 2009*; *Hsu et al., 2011*; *Yu et al., 2011*). One such potential mTOR substrate is IBTK. We noted that seven mTOR-regulated Ser/Thr sites are clustered in a short region of IBTK and hyperphosphorylation of IBTK$_{900-1150aa}$ mutant was abolished when mTOR activity was inhibited. In particular, using in vitro kinase assays combined with MS, we have identified S999/S1004 as sites phosphorylated by mTOR, and S990/993 as sites phosphorylated by S6K1. However, the kinases responsible for the remaining three phosphorylation sites are still unidentified, necessitating further investigation. Understanding the precise mechanisms by which mTOR regulates IBTK phosphorylation and activity could offer insights into the broader functions of mTOR signaling in protein synthesis and cellular homeostasis.

## Materials and methods

### Cell lines and lentiviral infection

SiHa, HeLa, 293T, H1299, and CT26 cells were acquired from American Type Culture Collection (ATCC). All cell lines were cultured in Dulbecco's modified Eagle's medium (DMEM) supplemented with 10% fetal bovine serum (FBS) and 1% penicillin/streptomycin at 37°C with 5% $CO_2$. We routinely perform DNA fingerprinting and PCR to verify the authenticity of the cell lines and to ensure they are free of mycoplasma infection. We conducted transient transfection using EZ Trans (Shanghai Life-iLab Biotech). For lentiviral transfection, we transfected pLKO shRNA KD plasmids and virus packing constructs into 293T cells. The viral supernatant was collected after 48 hr. SiHa, HeLa, or CT26 cells were then infected with the viral supernatant in the presence of polybrene (8 μg/ml) and selected in growth media containing puromycin (1.5 μM). The gene-specific shRNA sequences can be found in *Supplementary file 1f*.

### Antibodies, recombinant proteins, and chemicals

The information of antibodies, recombinant proteins, and chemicals used in this study is listed in *Supplementary file 1g and h*.

## Gene KO cell lines generation

The sgRNAs targeting the IBTK gene were designed using an online CRISPR design tool (http://crispr.mit.edu) and subcloned into the pX459 construct from Dr. Feng Zhang's lab. SiHa, 293T, and H1299 cells were plated and transfected with pX459 constructs overnight. After 24 hr of transfection, cells were exposed to puromycin (1.5 μM) for 1 week. The surviving cells were then seeded in a 96-well plate by limited dilution to isolate monoclonal cell lines. After 10 days, KO cells were screened using western blot (WB) analysis and validated through Sanger sequencing. *Supplementary file 1f* lists the sequences of gene-specific sgRNAs.

## DOX-inducible expression and protein complex purification

To generate stable cell lines with inducible IBTK gene expression, Flp-In T-REx 293 cells were co-transfected with pOG44 and pcDNA5-FLAG-BirA*-IBTK constructs. After 2 days of transfection, the cells were selected with hygromycin (100 μg/ml) for 2 weeks, and then the positive clones were pooled and amplified. To induce exogenous IBTK expression, DOX (10 ng/ml) was added to the stable cell lines. For purification in the AP-MS pipeline, the sample was lysed in 3 ml (for each plate) of NP-40 lysis buffer (0.1% NP-40) containing fresh protease inhibitor and kept on ice for 2 hr. The homogenate was centrifuged for 30 min at 12,000 rpm at 4°C. Cleared lysates were filtered through 0.45 μM spin filters (Millipore) and immunoprecipitated with anti-FLAG antibody-conjugated M2 agarose (Sigma). The bound polypeptides were eluted with the FLAG peptide (Sigma). For BioID approach, the cell pellet was thawed in 3 ml of BC100 lysis buffer (20 mM Tris-Cl (pH 7.9), 100 mM NaCl, 0.2 mM EDTA, 20% glycerol) containing fresh protease inhibitor and kept on ice for 2 hr. The homogenate was centrifuged for 30 min at 12,000 rpm at 4°C. Cleared lysates were filtered through 0.45 μM spin filters (Millipore) and immunoprecipitated with Strep-Tactin beads (IBA Lifescience). The bound polypeptides were eluted with 50 nM biotin. Finally, the eluates were resolved by SDS-PAGE for Coomassie Blue staining. Gel bands were cut out and subjected to MS sequencing.

## Protein half-life assays

To measure the protein half-life, we added cycloheximide (100 μg/ml) to the media of parental and IBTK-KO SiHa cells. At specific time points thereafter, we prepared whole cell lysates (WCLs) and detected them through WB analysis using the specified antibodies.

## In vivo ubiquitination assays

We transfected 293T cells with HA-ubiquitin and other specified constructs. After 36 hr, we harvested the cells and lysed them in a 1% SDS buffer (20 mM Tris-Cl (pH 7.4), 0.5 mM EDTA, and 1 mM DTT). The lysate was then boiled at 105°C for 10 min. Next, we added Strep-Tactin beads (IBA Lifescience) and a 10-fold volume of lysis buffer (20 mM Tris-Cl [pH 7.4], 100 mM NaCl, 0.2 mM EDTA, 0.5% NP-40, and 1× protease inhibitor cocktail) to the lysate, and incubated it overnight at 4°C with shaking. The resulting mixture was washed four times with BC100 buffer and eluted with biotin (50 mM) for 1 hr at 4°C. Finally, we detected the ubiquitinated substrates through WB analysis using an anti-HA antibody.

## In vitro ubiquitination assays

In vitro ubiquitination assays were performed using a protocol reported previously with some modifications (*Theurillat et al., 2014*). Briefly, 2 μg of APP-BP1/Uba3, 2 μg of His-UBE2M, and 5 μg of NEDD8 were incubated at 30°C for 2 hr in the presence of ATP. The thioester-loaded His-UBE2M-NEDD8 was further incubated with 3 μg of His-DCNL2 and 6 μg of CUL3-RBX1 at 4°C for 2 hr to obtain neddylated CUL3-RBX1. The neddylated CUL3-RBX1, 5 μg of FLAG-IBTK which was immunoprecipitated from 293T cells, 5 μg of ubiquitin, 500 ng of E1 enzyme, 750 ng of E2 enzyme (UbcH5a and UbcH5b), and 5 μg of GST-eIF4A1 were incubated with 0.6 μl of 100 mM ATP, 1.5 μl of 20 μM ubiquitin, 3 μl of 10× ubiquitin reaction buffer (500 mM Tris-Cl [pH 7.5], 50 mM KCl, 50 mM NaF, 50 mM MgCl$_2$, and 5 mM DTT), 3 μl of 10× energy regeneration mix (200 mM creatine phosphate and 2 μg/μl creatine phosphokinase), and 3 μl of 10× protease inhibitor cocktail at 30°C for 2 hr, followed by WB analysis.

## IF and confocal microscopy

For IF, SiHa cells were seeded on glass coverslips in 24-well plates and harvested at 80% confluence. The cells were washed with phosphate-buffered saline (PBS) and fixed with 4% paraformaldehyde

(PFA) in PBS. After permeabilization with 0.4% Triton X-100 for 10 min and then in the blocking solution (PBS plus 5% donkey serum), for 30 min at room temperature (RT). The cells were then incubated with primary antibodies at 4°C overnight. After washing with PBST buffer, fluorescence-labeled secondary antibodies were applied. DAPI was utilized to stain nuclei. The glass coverslips were mounted on slides and imaged using a confocal microscope (LSM880, Zeiss) with a 63*/1.4NA Oil PSF Objective. Quantitative analyses were performed using ImageJ software.

For xenograft tumor tissue staining, the tumor tissues were isolated from mice after perfusion with 0.1 M PBS (pH 7.4) and fixed for 3 days with 4% PFA at 4°C. The tumor tissues were then placed in 30% sucrose solution for 2 days for dehydration. The tumors were embedded into the OCT block and frozen for cryostat sectioning. Cryostat sections (45 µm thick) were washed with PBS, and then incubated in blocking solution (PBS containing 10% goat serum, 0.3% Triton X-100, pH 7.4) for 2 hr at RT. In antibody reaction buffer (PBS plus 10% goat serum, 0.3% Triton X-100, pH 7.4), the samples were stained with primary antibodies against active CD8 (1:100) and Granzyme B (1:200) overnight at 4°C, followed by Alexa 488 and 647 secondary antibodies (1:2000) at RT for 3 hr. DAPI was used for nuclear staining. The sections were then sealed with an anti-fluorescence quencher. The samples were visualized and imaged using a confocal microscope (FV3000, Olympus) along the z-axis with a 40× objective. The number of CD8$^+$ T cells and the area of Granzyme B were quantified using ImageJ by computing corresponding positive staining area. The analyses were performed in eight different units.

## Measurement of protein synthesis

We used puromycin labeling to measure nascent protein synthesis. The cells were treated with puromycin (1.5 µM) for 10 min before being lysed by 1× SDS sample buffer. The samples were analyzed by WB using anti-puromycin antibody.

## Pull-down assays using m$^7$GTP-Sepharose

SiHa cells were washed in PBS and lysed in 1 ml NP-40 lysis buffer supplemented with protease inhibitor. For each sample, the WCLs were incubated with 50 µl 7-methyl-GTP Sepharose 4B beads (GE Healthcare) for 2 hr at 4°C. Then, the resins were washed five times with BC100 buffer, and the proteins bound to the washed Sepharose resins were lysed by using 1× SDS sample buffer and boiled at 105°C for 5 min, after which the lysates were subjected to SDS-PAGE followed by WB analysis with the indicated antibodies.

## Dual-luciferase assays

The bicistronic reporters, pH and pE, were kindly provided by Dr. Peter Sarnow (Stanford University). The cells were seeded onto 24-well plates and co-transfected with each reporter plasmid. After 24 hr of transfection, the cells were harvested in passive lysis buffer and immediately quick-frozen at –80°C to fully lyse them. Dual-luciferase assays were performed using a dual-luciferase reporter assay system following the manufacturer's instructions (Promega). The luciferase activity was measured using an Envision Multilabel Plate Reader (PerkinElmer). Three independent experiments were performed.

## RNA isolation and quantitative reverse transcription polymerase chain reaction

Total RNAs were isolated from cells using the TRIzol reagent (Thermo) following the manufacturer's instructions. Concentrations and purity of RNAs were determined by measuring the ultraviolet absorbance at 260 nm and 280 nm using a NanoDrop spectrophotometer (Thermo). cDNAs were reverse-transcribed using a HiScript III 1st Strand cDNA Synthesis Kit (Vazyme), followed by amplification of cDNA using ChamQ SYBR qPCR Master Mix (Vazyme). The relative mRNA levels of genes were quantified using the $2^{-\Delta\Delta Ct}$ method, with normalization to *Actin*. The primer sequences are listed in *Supplementary file 1f*.

## CCK-8 assays

The cell proliferation rates of HeLa or SiHa were determined using the Cell Counting Kit-8 (CCK-8) Kit (Beyotime) according to the manufacturer's instructions. Briefly, cells were seeded onto 96-well plates at a density of 1000 cells per well in triplicate. During a 0–5 or 6 day culture period, 10 µl of the

CCK-8 solution was added to cell culture, and incubated for 2 hr. Then, samples were measured at an absorbance of 450 nm using a microplate absorbance reader (Bio-Rad).

## Colony formation assays

HeLa or SiHa cells were seeded in six-well plates containing 1000 individual cells per well in triplicate. After incubating for 2 weeks, cells were fixed with 4% PFA for 15 min at 37°C and then subjected to Giemsa dye (Solarbio) staining for 20 min. Then, the cells were washed with water by dropping gently, and air-dried at RT. The number of colonies was photographed using a digital photo camera (Nikon) and quantified using ImageJ.

## Migration and invasion assays

Cell migration and invasion rates were determined using Transwell chamber (Corning). HeLa or SiHa cells were pre-cultured in serum-free media for 24 hr. In the case of migration assays, $2 \times 10^5$ cells were seeded in serum-free media in the upper chamber, and the lower chamber was filled with DMEM containing 5% FBS. After 48 hr, the cells were fixed with 4% PFA for 15 min at 37°C followed by staining with Giemsa dye (Solarbio) for 20 min. The non-migrating cells on the upper chambers were carefully removed with a cotton swab and migrated cells on the underside of the filter were observed and counted in three different fields. The protocol for invasion assays is similar to the migration assays, with the exception that Matrigel (Corning) is added to the upper chambers before seeding the cells. Three independent experiments were conducted.

## Sphere-formation assays

SiHa or HeLa cells suspension ($2 \times 10^5$ cells/well) were mixed with Matrigel and then plated in 24-well ultra-low attachment plates in DMEM containing 10% FBS. Fresh media were added every 3 days. The floating spheres that grew in 1–2 weeks were captured using a digital photo camera (Nikon), and their number and size were measured using ImageJ. Three independent experiments were conducted.

## Apoptosis detection assays

To measure the apoptosis rates of cells, annexin-V-FITC (fluorescein isothiocyanate) and propidium iodide double staining was employed. SiHa or HeLa cells were cultured in six-well plates for 12 hr, followed by the addition of silvestrol (100 nM) for the specified time period. After that, the suspension and adherent cells were collected separately and then combined for apoptosis detection using the Apoptosis Detection Kit (Dojindo). Similarly, Rocaglamide A (200 nM) was also administered to initiate apoptosis. All flow cytometric analyses were conducted using a flow cytometer (FACSCalibur, BD Biosciences). Three independent experiments were performed.

## Mouse tumor implantation assays

All experimental protocols were approved in advance by the Ethics Review Committee for Animal Experimentation of Shanghai First Maternity and Infant Hospital. 4-Week-old BALB/c female mice and athymic nude mice (SLAC Laboratory) were bred and maintained in our institutional pathogen-free mouse facilities. Subsequently, $1 \times 10^6$ parental or IBTK-KD SiHa cells were subcutaneously (s.c.) injected into female BALB/c mice aged 6–8 weeks. After 20 days of tumor cell injection, the mice were euthanized, and in vivo solid tumors were excised and weighed. The growth of tumors was measured every 5 days in two dimensions using a digital caliper, with tumor volumes calculated using the ellipsoid volume formula: $V = (L \times W^2)/2$, where L is the length and W is the width. A portion of tumors were fixed in formalin and embedded in paraffin for IF analysis. In a similar procedure, $1 \times 10^7$ parental and IBTK-KD HeLa cells were s.c. injected into nude mice. After 20 days of tumor cell injection, the mice were euthanized, and in vivo solid tumors were excised and weighed.

## Recombinant protein production and in vitro kinase assays

The pGEX-4T-2 construct containing IBTK$_{900-1150aa}$ was transformed into *Escherichia coli* Rosetta (DE3) to express the recombinant GST-tagged proteins. These proteins underwent a two-step purification process on glutathione-agarose beads (Pharmacia), followed by affinity purification on Amicon Ultra-0.5 Centrifugal Filter Devices (Millipore). IBTK phosphorylation levels were assessed using in vitro kinase assays. Purified GST-tagged IBTK$_{900-1150aa}$ at 2 µg was mixed with kinases mTOR/mLST8 or S6K1

(Carna Biosciences) in a reaction mixture that contained 10× kinase buffer (CST), 10 mM ATP (Sigma) for 1 hr at 30°C. The reaction was terminated by adding 2× SDS loading buffer and boiling at 105°C for 5 min. SDS-PAGE was used to separate proteins, with gel bands cut out and subjected to MS sequencing.

## IHC for human CESC specimens

IBTK protein expression was detected using a tissue microarray (HUteS154Su01, Shanghai Outdo Biotech), which included 152 points comprising 117 CESC tissues and 35 adjacent normal cervical tissues. The TMA slide was first baked at 65°C for 30 min and then deparaffinized in xylene. The tissue sections were passed through graded alcohol before being subjected to antigen retrieval with 1 mM EDTA, pH 9.0 (Servicebio) in a microwave at 50°C for 10 min and 30°C for 10 min. Subsequently, the sections were treated with 3% $H_2O_2$ for 25 min to quench endogenous peroxidase activity and washed carefully in PBS (pH 7.4) thrice. A solution of 3% bovine serum albumin was added onto the slide to evenly cover the tissue, which was then incubated at 37°C for 30 min. Next, the slide was incubated with diluted antibodies overnight at 4°C. After rinsing with PBS for three times, the sections were treated with horseradish peroxidase-conjugated mouse antibody (Servicebio) for 50 min, followed by 3,3′-diaminobenzidine incubation. Finally, the slide was counterstained with 0.1% hematoxylin, dehydrated, and covered before visualization under a confocal microscope. Each sample was scored based on the intensity of staining. All IHC data were interpreted by the same qualified pathologist for consistency.

## Statistical analysis

Kaplan-Meier plots were used to generate survival curves, which displayed p values, fold changes, and ranks. The results of Kaplan-Meier plots were displayed with HR and p or Cox p values from a log-rank test. Band intensities of WB results were calculated by ImageJ in accordance with the manufacturer's instructions. Statistical analyses were performed using GraphPad Prism (GraphPad Software), and the differences between two groups were analyzed using Student's t-test while the differences between multiple groups were analyzed using one-way or two-way analysis of variance (ANOVA), unless otherwise specified. All data were displayed as means ± SD values for experiments conducted with at least three replicates.

## Acknowledgements

This work was in part supported by the National Natural Science Foundation of China (No. 82272992, 91954106, and 81872109 to KG; No. 92357301, 32370726, 91957125, 81972396 to CW), the State Key Development Programs of China (No. 2022YFA1104200 to CW), the Natural Science Foundation of Shanghai (No. 22ZR1449200 to KG; 22ZR1406600 to CW), and the Open Research Fund of State Key Laboratory of Genetic Engineering, Fudan University (No. SKLGE-2111 to KG). Science and Technology Research Program of Shanghai (No. 9DZ2282100). Open Research Fund of the Shanghai Key Laboratory of Maternal and Fetal Medicine (mfmkf202204 to CW).

## Additional information

### Funding

| Funder | Grant reference number | Author |
| --- | --- | --- |
| National Natural Science Foundation of China | 82272992 | Kun Gao |
| National Natural Science Foundation of China | 92357301 | Chenji Wang |
| State Key Development Programs of China | 2022YFA1104200 | Chenji Wang |
| National Science Foundation of Shanghai | 22ZR1449200 | Kun Gao |

| Funder | Grant reference number | Author |
| --- | --- | --- |
| National Science Foundation of Shanghai | 22ZR1406600 | Chenji Wang |
| Open Research Fund of State Key Laboratory of Genetic Engineering, Fudan University | No. SKLGE-2111 | Kun Gao |
| Science and Technology Research Program of Shanghai | 9DZ2282100 | Chenji Wang |
| Open Research Fund of the Shanghai Key Laboratory of Maternal and Fetal Medicine | mfmkf202204 | Chenji Wang |
| National Natural Science Foundation of China | 32370726 | Chenji Wang |
| National Natural Science Foundation of China | 91954106 | Kun Gao |
| National Natural Science Foundation of China | 81872109 | Kun Gao |
| National Natural Science Foundation of China | 91957125 | Chenji Wang |
| National Natural Science Foundation of China | 81972396 | Chenji Wang |

The funders had no role in study design, data collection and interpretation, or the decision to submit the work for publication.

## Author contributions

Dongyue Jiao, Huiru Sun, Data curation, Formal analysis, Validation, Investigation, Visualization, Methodology; Xiaying Zhao, Data curation, Formal analysis, Validation, Visualization, Methodology; Yingji Chen, Zeheng Lv, Qing Shi, Validation, Investigation; Yao Li, Resources, Supervision, Project administration; Chenji Wang, Conceptualization, Resources, Data curation, Formal analysis, Supervision, Funding acquisition, Validation, Investigation, Visualization, Methodology, Writing - original draft, Project administration, Writing - review and editing; Kun Gao, Conceptualization, Resources, Supervision, Funding acquisition, Validation, Investigation, Visualization, Methodology, Writing - original draft, Project administration, Writing - review and editing

## Author ORCIDs

Dongyue Jiao http://orcid.org/0009-0002-4061-0578
Xiaying Zhao http://orcid.org/0009-0002-4067-9655
Zeheng Lv http://orcid.org/0009-0009-6355-9374
Chenji Wang http://orcid.org/0000-0002-5752-6439

## Ethics

All participants gave written informed consent to participate and to have their data published in scientific journals. This study involved human subjects received ethical approval from approved by the Ethics Review Committee of Shanghai First Maternity and Infant Hospital (Permit Number: KS2281).

All procedures for animal care and animal experiments were carried out in accordance with the guidelines of the Care and Use of Laboratory Animals proposed by School of Medicine, Tongji University (Permit Number: TJBG10022102).

Reviewer #1 (Public Review): https://doi.org/10.7554/eLife.92236.3.sa1
Reviewer #2 (Public Review): https://doi.org/10.7554/eLife.92236.3.sa2
Author response https://doi.org/10.7554/eLife.92236.3.sa3

## Additional files

### Supplementary files

• Supplementary file 1. MS data and the information of sequences and reagents. (a) High-confidence inhibitor of Bruton's tyrosine kinase (IBTK)-interacting protein identified with affinity purification coupled with mass spectrometry (AP-MS) method. (b) High-confidence IBTK-interacting protein identified with biotinylation identification (BioID) method. (c) Ubiquitinated peptide sequences of eIF4A1. (d) Phosphorylated peptide sequences of IBTK by mTOR. (e) Phosphorylated peptide sequences of IBTK by S6K1. (f) Sequence information. (g) Antibody information. (h) Cell lines, cell cultures, chemicals, and kits.

• MDAR checklist

### Data availability

The mass spectrometry proteomics data have been deposited to the ProteomeXchange Consortium via the PRIDE partner repository with the dataset identifier PXD039031, PXD038307, and PXD038286.

The following datasets were generated:

| Author(s) | Year | Dataset title | Dataset URL | Database and Identifier |
|---|---|---|---|---|
| Jiao D, Sun H, Chen Y, Lv Z, Li Y, Wang C, Gao K | 2023 | The identification of ubiquitination sites of IBTK by shotgun proteomics | https://www.ebi.ac.uk/pride/archive/projects/PXD039031/ | PRIDE, PXD039031 |
| Jiao D, Sun H, Zhao X, Chen Y, Lv Z, Shi Q, Wang C, Gao K, Li Y | 2023 | The identification of phosphorylation sites of IBTK by shotgun proteomics | https://www.ebi.ac.uk/pride/archive/projects/PXD038307 | PRIDE, PXD038307 |
| Jiao D, Sun H, Zhao X, Chen Y, Lv Z, Shi Q, Li Y, Wang C, Gao K | 2023 | Define the interactome of IBTK by Affnity purification coupled with mass spectrometry (AP-MS) and proximity-dependent biotinylaion identification (BioID) methods | https://www.ebi.ac.uk/pride/archive/projects/PXD038286 | PRIDE, PXD038286 |

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
