## [Editor Report · eLife assessment]

This study reports a novel substrate and a mediator of oncogenesis downstream of mTORC1, a **fundamental** advance in our understanding of the mechanistic basis of mTORC1-regulated cap-dependent translation and protein synthesis. Using an array of biochemical, proteomic and functional assays, the authors provide **compelling** evidence for a novel mTORC1/S6K1-IBTK-eIF4A1 signaling axis that promotes cancer pathogenic translation. This work is of broad interest and significance, given the importance of aberrant protein synthesis in cancer.

---

## [Referee Report · Reviewer #1 (Public Review)]

In this study, the authors examined the role of IBTK, a substrate-binding adaptor of the CRL3 ubiquitin ligase complex, in modulating the activity of the eiF4F translation initiation complex. They find that IBTK mediates the non-degradative ubiquitination of eiF4A1, promotes cap-dependent translational initiation, nascent protein synthesis, oncogene expression, and tumor cell growth. Correspondingly, phosphorylation of IBTK by mTORC1/ S6K1 increases eIF4A1 ubiquitination and sustains oncogenic translation.

Strengths:

This study utilizes multiple biochemical, proteomic, functional and cell biology assays to substantiate their results. Importantly, the work nominates IBTK as a unique substrate of mTORC1, and further validates eiF4A1 ( a crucial subunit of the ei44F complex) as a promising therapeutic target in cancer. Since IBTK interacts broadly with multiple members of the translational initial complex- it will be interesting to examine its role in eiF2alpha-mediated ER stress as well as eiF3-mediated translation. Additionally, since IBTK exerts pro-survival effects in multiple cell types, it will be of relevance to characterize the role of IBTK in mediating increased mTORC1 mediated translation in other tumor types, thus potentially impacting their treatment with eiF4F inhibitors.

Limitations/Weaknesses:

The findings are mostly well supported by data, but some areas need clarification and could potentially be enhanced with further experiments:

(1) Since eiF4A1 appears to function downstream of IBTK1, can the effects of IBTK1 KO/KD in reducing puromycin incorporation ( in Fig 3A), cap-dependent luciferase reporter activity (Fig 3G), reduced oncogene expression ( Fig 4A) or 2D growth/ invasion assays (Fig 4) be overcome or bypassed by overexpressing eiF4A1? These could potentially be tested in future studies.

(2) The decrease in nascent protein synthesis in puromycin incorporation assays in Figure 3A suggests that the effects of IBTK KO are comparable to and additive with silvesterol. It would be of interest to examine whether silvesterol decreases nascent protein synthesis or increases stress granules in the IBTK KO cells stably expressing IBTK as well.

(3) The data presented in Figure 5 regarding the role of mTORC1 in IBTK-mediated eiF4A1 ubiquitination needs further clarification on several points:

- It is not clear if the experiments in Figure 5F with Phos-tag gels are using the FLAG-IBTK deletion mutant or the peptide containing the mTOR sites as it is mentioned on line 517, page 19 "To do so, we generated an IBTK deletion mutant (900-1150 aa) spanning the potential mTORC1-regulated phosphorylation sites" This needs further clarification.

-It may be of benefit to repeat the Phos tag experiments with full length FLAG-IBTK and/or endogenous IBTK with molecular weight markers indicating size of migrated bands.

-Additionally, torin or Lambda phosphatase treatment may be used to confirm the specificity of the band in separate experiments.

-Phos-tag gels with the IBTK CRISPR KO line would also help confirm that the non-phosphorylated band is indeed IBTK.

-It is unclear why the lower, phosphorylated bands seem to be increasing ( rather than decreasing) with AA starvation/ Rapa in Fig 5H.

---

## [Referee Report · Reviewer #2 (Public Review)]

Summary:

This study by Sun et al. identifies a novel role for IBTK in promoting cancer protein translation, through regulation of the translational helicase eIF4A1. Using a multifaceted approach, the authors demonstrate that IBTK interacts with and ubiquitinates eIF4A1 in a non-degradative manner, enhancing its activation downstream of mTORC1/S6K1 signaling. This represents a significant advance in elucidating the complex layers of dysregulated translational control in cancer.

Strengths:

A major strength of this work is the convincing biochemical evidence for a direct regulatory relationship between IBTK and eIF4A1. The authors utilize affinity purification and proximity labeling methods to comprehensively map the IBTK interactome, identifying eIF4A1 as a top hit. Importantly, they validate this interaction and the specificity for eIF4A1 over other eIF4 isoforms by co-immunoprecipitation in multiple cell lines. Building on this, they demonstrate that IBTK catalyzes non-degradative ubiquitination of eIF4A1 both in cells and in vitro through the E3 ligase activity of the CRL3-IBTK complex. Mapping IBTK phosphorylation sites and showing mTORC1/S6K1-dependent regulation provides mechanistic insight. The reduction in global translation and eIF4A1-dependent oncoproteins upon IBTK loss, along with clinical data linking IBTK to poor prognosis, support the functional importance. Finally, the impact of IBTK on eIF4A1 target gene expression in colon and lung cancer cell lines, strengthens these findings.

Weaknesses:

While the effects of IBTK knockout/over-expression on bulk protein synthesis are shown, the expression of several eIF4A1 target oncogenes remains unchanged.

Summary:

Overall, this study significantly advances our understanding of how aberrant mTORC1/S6K1 signaling promotes cancer pathogenic translation via IBTK and eIF4A1. The proteomic, biochemical and phosphorylation mapping approaches established here provide a blueprint for interrogating IBTK function. These data should galvanize future efforts to target the mTORC1/S6K1-IBTK-eIF4A1 axis as an avenue for cancer therapy, particularly in combination with eIF4A inhibitors.

---

## [Author Response]

The following is the authors’ response to the original reviews.

**Reviewer #1 (Public Review):**
In this study, the authors examined the role of IBTK, a substrate-binding adaptor of the CRL3 ubiquitin ligase complex, in modulating the activity of the eiF4F translation initiation complex. They find that IBTK mediates the non-degradative ubiquitination of eiF4A1, promotes cap-dependent translational initiation, nascent protein synthesis, oncogene expression, and tumor cell growth. Correspondingly, phosphorylation of IBTK by mTORC1/ S6K1 increases eIF4A1 ubiquitination and sustains oncogenic translation.Strengths:This study utilizes multiple biochemical, proteomic, functional, and cell biology assays to substantiate their results. Importantly, the work nominates IBTK as a unique substrate of mTORC1, and further validates eiF4A1 (a crucial subunit of the ei44F complex) as a promising therapeutic target in cancer. Since IBTK interacts broadly with multiple members of the translational initial complex - it will be interesting to examine its role in eiF2alpha-mediated ER stress as well as eiF3-mediated translation. Additionally, since IBTK exerts pro-survival effects in multiple cell types, it will be of relevance to characterize the role of IBTK in mediating increased mTORC1 mediated translation in other tumor types, thus potentially impacting their treatment with eiF4F inhibitors.Limitations/Weaknesses:The findings are mostly well supported by data, but some areas need clarification and could potentially be enhanced with further experiments:(1) Since eiF4A1 appears to function downstream of IBTK1, can the effects of IBTK1 KO/KD in reducing puromycin incorporation (in Fig 3A), cap-dependent luciferase reporter activity (Fig 3G), reduced oncogene expression (Fig 4A) or 2D growth/ invasion assays (Fig 4) be overcome or bypassed by overexpressing eiF4A1? These could potentially be tested in future studies.

We appreciate the reviewer for bringing up this crucial point. As per the reviewer's suggestion, we conducted experiments where we overexpressed Myc-eIF4A1 in IBTK-KO SiHa cells. Our findings indicate that increasing levels of eIF4A1 through ectopic overexpression is unable to reverse the decrease in puromycin incorporation (Fig. S3C) and protein expression of eIF4A1 targets caused by IBTK ablation (Fig. S4E). These results clearly demonstrate that IBTK ablation-induced eIF4A1 dysfunctions cannot be rescued by simply elevating eIF4A1 protein levels. Given the above results are negative, the impacts of eIF4A1 overexpression on the 2D growth/invasion capacities of IBTK-KO cells were not further examined. We sincerely appreciate the reviewer's understanding regarding this matter.

(2) The decrease in nascent protein synthesis in puromycin incorporation assays in Figure 3A suggest that the effects of IBTK KO are comparable to and additive with silvesterol. It would be of interest to examine whether silvesterol decreases nascent protein synthesis or increases stress granules in the IBTK KO cells stably expressing IBTK as well.

We appreciate the reviewer for bringing up this crucial point. We have showed that silvestrol treatment still decreased nascent protein synthesis in IBTK-KO cells overexpressing FLAG-IBTK as well (Fig. S3B).

(3) The data presented in Figure 5 regarding the role of mTORC1 in IBTK- mediated eiF4A1 ubiquitination needs further clarification on several points:It is not clear if the experiments in Figure 5F with Phos-tag gels are using the FLAG-IBTK deletion mutant or the peptide containing the mTOR sites as it is mentioned on line 517, page 19 "To do so, we generated an IBTK deletion mutant (900-1150 aa) spanning the potential mTORC1-regulated phosphorylation sites" This needs further clarification.

We appreciate the reviewer for bringing up this crucial point. The IBTK deletion mutant used in Fig. 5F is FLAG-IBTK900-1150aa. We have annotated it with smaller font size in the panel (red box) in Author response image 1.

It may be of benefit to repeat the Phos tag experiments with full-length FLAG- IBTK and/or endogenous IBTK with molecular weight markers indicating the size of migrated bands.

We appreciate the reviewer for bringing up this crucial point. We attempted to perform Phos-tag assays to detect the overexpressed full-length FLAG-IBTK or endogenous IBTK. However, we encountered difficulties in successfully transferring the full-length FLAG-IBTK or endogenous IBTK onto the nitrocellulose membrane during Phos-tag WB analysis. This is likely due to the limitations of this technique. Based on our experience, phos-tag gel is less efficient in detecting protein motility shifts with large molecular weights. As the molecular weight of IBTK protein is approximately 160 kDa, it falls within this category. Considering these technical constraints, we did not include Phos-tag assay results for full-length IBTK in our study. We sincerely appreciate the reviewer's understanding regarding this matter.

The binding of Phos-tag to phosphorylated proteins induces a mobility shift during gel electrophoresis or protein separation techniques. This shift allows for the visualization and quantification of phosphorylated proteins separately from non-phosphorylated proteins. It's important to note that these mobility shifts indicate phosphorylation status, rather than actual molecular weights. pre- stained protein markers are typically used as a reference to assess the efficiency of protein transfer onto the membrane [Ref: 1]. Considering the aforementioned reasons, we did not add molecular weights to the WB images.

Reference [1]. FUJIFILM Wako Pure Chemical Corporation, https://www.wako- chemicals.de/media/pdf/c7/5e/20/FUJIFILM-Wako_Phos-tag-R.pdf

Additionally, torin or Lambda phosphatase treatment may be used to confirm the specificity of the band in separate experiments.

We appreciate the reviewer for bringing up this crucial point. Torin1 is a synthetic mTOR inhibitor by preventing the binding of ATP to mTOR, leading to the inactivation of both mTORC1 and mTORC2, whereas rapamycin primarily targets mTORC1 activity and may inhibit mTORC2 in certain cell types after a prolonged treatment. We have identified that the predominant mediator of IBTK phosphorylation is the mTORC1/S6K1 complex. Therefore, in this context, we think that rapamycin is sufficient to inactivate the mTORC1/S6K1 pathway. As shown in Fig. 5F, the phosphorylated IBTK900-1150aa was markedly decreased while the non-phosphorylated form was simultaneously increased in rapamycin- treated cells. As per the reviewer's suggestion, we treated FLAG-IBTK900-1150aa overexpressed cells with lambda phosphatase. As shown in Fig. 5G, lambda phosphatase treatment completely abolished the mobility shifts of phosphorylated FLAG-IBTK900-1150aa. Additionally, the lowest band displayed an abundant accumulation of the non-phosphorylated form of FLAG-IBTK900-1150aa. These findings confirm that the mobility shifts observed in WB analysis correspond to the phosphorylated forms of FLAG-IBTK900-1150aa.

Phos-tag gels with the IBTK CRISPR KO line would also help confirm that the non-phosphorylated band is indeed IBTK.

We appreciate the reviewer for bringing up this crucial point. As we state above, we performed Phos-tag assays to detect the mobility shifts of phosphorylated FLAG-IBTK900-1150aa. Anti-FLAG antibody, but not the anti-IBTK antibody was used for WB detection. This antibody does not exhibit cross-reactivity with endogenous IBTK.

It is unclear why the lower, phosphorylated bands seem to be increasing (rather than decreasing) with AA starvation/ Rapa in Fig 5H.

We appreciate the reviewer for bringing up this crucial point. We think the panel the reviewer mentioned is Fig. 5F. According to the principle of Phos-tag assays, proteins with higher phosphorylation levels have slower migration rates on SDS-PAGE, while proteins with lower phosphorylation levels have faster migration rates.

As shown in Author response image 2, the green box indicates the most phosphorylated forms of FLAG-IBTK900-1150aa, the red box indicates the moderately phosphorylated forms of FLAG-IBTK900-1150aa, and the yellow box indicates the non-phosphorylated forms of FLAG-IBTK900-1150aa. AA starvation or Rapamycin treatment reduced the hyperphosphorylated forms of FLAG-IBTK900-1150aa (green box), while simultaneously increasing the hypophosphorylated (red box) and non- phosphorylated (yellow box) forms of FLAG-IBTK900-1150aa. Thus, we conclude that AA starvation or Rapamycin treatment leads to a marked decrease in the phosphorylation levels of FLAG-IBTK900-1150aa.

**Author response image 2. sa3fig2:** 

**Reviewer #2 (Public Review):**
Summary:This study by Sun et al. identifies a novel role for IBTK in promoting cancer protein translation, through regulation of the translational helicase eIF4A1. Using a multifaceted approach, the authors demonstrate that IBTK interacts with and ubiquitinates eIF4A1 in a non-degradative manner, enhancing its activation downstream of mTORC1/S6K1 signaling. This represents a significant advance in elucidating the complex layers of dysregulated translational control in cancer.Strengths:A major strength of this work is the convincing biochemical evidence for a direct regulatory relationship between IBTK and eIF4A1. The authors utilize affinity purification and proximity labeling methods to comprehensively map the IBTK interactome, identifying eIF4A1 as a top hit. Importantly, they validate this interaction and the specificity for eIF4A1 over other eIF4 isoforms by co- immunoprecipitation in multiple cell lines. Building on this, they demonstrate that IBTK catalyzes non-degradative ubiquitination of eIF4A1 both in cells and in vitro through the E3 ligase activity of the CRL3-IBTK complex. Mapping IBTK phosphorylation sites and showing mTORC1/S6K1-dependent regulation provides mechanistic insight. The reduction in global translation and eIF4A1- dependent oncoproteins upon IBTK loss, along with clinical data linking IBTK to poor prognosis, support the functional importance.Weaknesses:While these data compellingly establish IBTK as a binding partner and modifier of eIF4A1, a remaining weakness is the lack of direct measurements showing IBTK regulates eIF4A1 helicase activity and translation of target mRNAs. While the effects of IBTK knockout/overexpression on bulk protein synthesis are shown, the expression of multiple eIF4A1 target oncogenes remains unchanged.Summary:Overall, this study significantly advances our understanding of how aberrant mTORC1/S6K1 signaling promotes cancer pathogenic translation via IBTK and eIF4A1. The proteomic, biochemical, and phosphorylation mapping approaches established here provide a blueprint for interrogating IBTK function. These data should galvanize future efforts to target the mTORC1/S6K1-IBTK-eIF4A1 axis as an avenue for cancer therapy, particularly in combination with eIF4A inhibitors.
**Reviewer #1 (Recommendations For The Authors):**
(1) Certain references should be provided for clarity. For e.g.,: Page 15, line 418 " The C-terminal glycine glycine (GG) amino acid residues are essential for Ub conjugation to targeted proteins".

We appreciate the reviewer for bringing up this crucial point. We have taken two fundamental review papers (PMID: 22524316, 9759494) on the ubiquitin system as references in this sentence.

(2) Please describe the properties of the ΔBTB mutant on page 15 when first describing it. What motifs does it lack and has it been described before in functional studies?

We appreciate the reviewer for bringing up this crucial point. We added a sentence to describe the properties of the ΔBTB mutant. This mutant lacks the BTB1 and BTB2 domains (deletion of aa 554–871), which have been previously demonstrated to be essential for binding to CUL3. The original reference has been added to the revised manuscript.

(3) In Figure 2G how do the authors explain the fact that co-expression of the Ub K-ALLR mutant, which is unable to form polyubiquitin chains, formed only a moderate reduction in IBTK-mediated eIF4A1 ubiquitination?

We appreciate the reviewer for bringing up this crucial point. The Ub K-ALLR mutant can indeed conjugate to substrate proteins, but it cannot form chains due to its absence of lysine residues, resulting in mono-ubiquitination. Multi- mono-ubiquitination refers to the attachment of single ubiquitin molecules to multiple lysine residues on a substrate protein. It's worth noting that a poly- ubiquitinated protein and a multi-mono-ubiquitinated protein appear strikingly similar in Western blot. Our findings demonstrated that the co-expression of the Ub K-ALL-R mutant resulted in only a modest reduction in IBTK-mediated eIF4A1 ubiquitination (Fig. 2G), and that eIF4A1 was ubiquitinated at twelve lysine residues when co-expressed with IBTK (Fig. S2F). As such, we conclude that the CRL3IBTK complex primarily catalyzes multi-mono-ubiquitination on eIF4A1..

(4) In Figure 5, The identity of the seven sites in the IBTK 7ST A mutants should be specified.

We appreciate the reviewer for bringing up this crucial point. We have specified the seven mutation sites in the IBTK-7ST A mutant (Fig. 6A).

(5) In Figure 5, the rationale for generating antibodies only to S990/992/993, as opposed to the other mTORC1/S6K motifs should be specified.

We appreciate the reviewer for bringing up this crucial point. Upon demonstrating that IBTK can be phosphorylated—with evidence from positive Phos-tag and in vitro phosphorylation assays—we sought to directly detect changes in the phosphorylation levels using an antibody specific to IBTK phosphorylation. However, the expense of generating seven phosphorylation- specific antibodies for each site is significant. Recognizing that S990/992/993 are three adjacent sites, we deemed it appropriate to generate a single antibody to recognize the phospho-S990/992/993 epitope. Moreover, out of the seven phosphorylation sites, S992 perfectly matches the consensus motif for S6K1 phosphorylation (RXRXXS). Utilizing this antibody allowed us to observe a substantial decrease in the phosphorylation levels of these three adjacent Ser residues in IBTK following either AA deprivation or Rapamycin treatment (Fig. 5L). We have specified these points in the manuscript.

**Reviewer #2 (Recommendations For The Authors):**
The following suggestions would strengthen the study:(1) Directly examine the effects of IBTK modulation (knockdown/knockout/ overexpression) on eIF4A1 helicase activity.

We appreciate the reviewer for bringing up this crucial point. We agree with the reviewer's suggestion that evaluating IBTK's influence on eIF4A1 helicase activity directly would enhance the strength of our conclusion. However, the current eIF4A1 helicase assays, as described in previous publications [Ref: 1, 2], can only be conducted using in vitro purified recombinant proteins. For instance, it is feasible to assess the varying levels of helicase activity exhibited by recombinant wild-type or mutant EIF4A1 proteins [Ref: 2]. Importantly, there is currently no reported methodology for evaluating the helicase activity of EIF4A1 in vivo, as mentioned by the reviewer in gene knockdown, knockout, or overexpression cellular contexts. Therefore, we have not performed these assays and we sincerely appreciate the reviewer's understanding in this regard. We sincerely appreciate the reviewer's understanding regarding this matter.

Reference:

[1] Chu J, Galicia-Vázquez G, Cencic R, Mills JR, Katigbak A, Porco JA, Pelletier J. CRISPR-mediated drug-target validation reveals selective pharmacological inhibition of the RNA helicase, eIF4A. Cell reports. 2016 Jun 14;15(11):2340-7.

[2] Chu J, Galicia-Vázquez G, Cencic R, Mills JR, Katigbak A, Porco JA, Pelletier J. CRISPR-mediated drug-target validation reveals selective pharmacological inhibition of the RNA helicase, eIF4A. Cell reports. 2016 Jun 14;15(11):2340-7.

(2) Justify why the expression of some but not all eIF4A1 target oncogenes is affected in IBTK-depleted/overexpressing cells. This is important if IBTK should be considered as a therapeutic target. The authors should consider which of the eIF4A1 targets are most impacted by IBTK KO. This would provide a more focused therapeutic approach in the future.

We appreciate the reviewer for bringing up this crucial point. As the reviewer has pointed out, we assessed the protein levels of ten reported eIF4A1 target genes across three cancer cell lines (Fig.4, Fig. S4A, C). We observed that IBTK depletion led to a substantial reduction in the protein levels of most eIF4A1- regulated oncogenes upon IBTK depletion, although there were some exceptions. For instance, IBTK KO in H1299 cells exerted minimal influence on the protein levels of ROCK1 (Fig. S4A). Several possible explanations might account for this observation: firstly, given that our list of eIF4A1 target genes collected from previous studies conducted using distinct cell lines, it is not unexpected for different lines to exhibit subtle differences in regulation of eIF4A1 target genes. Secondly, as a CRL3 adaptor, IBTK potentially performs other biological functions via ubiquitination of specific substrates; dysregulation of these could buffer the impact of IBTK KO on the protein expression of some eIF4A1 target genes. We added these comments to the Discussion section of the revised manuscript.

(3) Expand mTOR manipulation experiments (inhibition, Raptor knockout, activation) and evaluate impacts on IBTK phosphorylation, eIF4A1 ubiquitination, and translation.

The mTORC1 signaling pathway is constitutively active under normal culture conditions. In order to inhibit mTORC1 activation, we employed several approaches including AA starvation, Rapamycin treatment, or Raptor knockout. Our results have demonstrated that both AA starvation and rapamycin treatment led to a reduction in eIF4A1 ubiquitination (Fig. 5M). Moreover, we have included new findings in the revised manuscript, which highlight that Raptor knockout specifically decreases eIF4A1 ubiquitination (Fig. 5N). It is worth mentioning that the impacts of mTOR inhibition or activation on protein translation have been extensively investigated and documented in numerous studies. Therefore, in our study, we did not feel it necessary to examine these treatments further.

(4) Although not absolutely necessary, it would be nice to see if some of these findings are true in other cancer cell types.

We appreciate the reviewer for bringing up this crucial point. We concur with the reviewer's suggestion that including data from other cancer cell types would enhance the strength of our conclusion. While the majority of our data is derived from two cervical cancer cell lines, we have corroborated certain key findings— such as the impact of IBTK on eIF4A1 and its target gene expression—in H1299 cells (human lung cancer) (Fig. 2C, Fig. S4A, B) and in CT26 cells (murine colon adenocarcinoma) (Fig. S4C, D). Additionally, we demonstrated that IBTK promotes IFN-γ-induced PD-L1 expression and tumor immune escape in both the H1299 and CT26 cells (Fig. S6A-K).